# Neighbor-aware Geodesic Transportation for Neighborhood Refinery

## Abstract

Neighborhood refinery aims to enhance the neighbor relationships by refining the original distance matrix to ensure pairwise consistency. Traditional context-based methods, which encode instances alongside their local neighbors in a contextual affinity space, are limited in capturing global relationships and are vulnerable to the negative impacts of outliers in the neighborhood. To overcome these limitations, we propose a novel Neighbor-aware Geodesic Transportation (NGT) for the neighborhood refinery. NGT first constructs a global-aware distribution for each instance, capturing the intrinsic manifold relationships among all instances. This is followed by an optimization transportation process that utilizes the global-aware distribution within the underlying manifold, incorporating global geometric spatial information to generate a refined distance. NGT first involves Manifold-aware Neighbor Encoding (MNE) to project each instance into a global-aware distribution by constraining pairwise similarity with the corresponding affinity graph to capture global relationships. Subsequently, a Regularized Barycenter Refinery (RBR) module is proposed to integrate local neighbors into a barycenter, employing a Wasserstein term to reduce the influence of outliers. Lastly, Geodesic Transportation (GT) leverages geometric and global context information to transport the barycenter distribution along the geodesic paths within the affinity graph. Extensive evaluations on several tasks, such as re-ranking and deep clustering, demonstrate the superiority of our proposed NGT.

## 1 Introduction

*Neighborhood Refinery* (Iscen et al., 2017; Bai et al., 2019; Shao et al., 2023; Chen et al., 2024; Yu et al., 2023; Luo et al., 2024; Liu et al., 2019) is a critical task in machine learning and computer vision, which refines the similarity or distance among neighborhood samples by leveraging the underlying manifold structure. Specifically, given an initial distance matrix obtained from Euclidean space, neighborhood refinery adjusts the pairwise similarity by ensuring consistent relationships among instances, resulting in a more informative refined distance matrix. Due to the higher coherence exhibited by the neighborhood samples obtained from the refined matrix, neighborhood refinery can be effectively applied to image retrieval tasks (Lee et al., 2022; Radenović et al., 2019; Yang et al., 2021; Tolias et al., 2016) and self-supervised framework (Dwibedi et al., 2021; Koohpayegani et al., 2021; Van Gansbeke et al., 2020; Niu et al., 2022) for re-ranking and deep clustering, *e.g.*, the refined query-to-gallery similarity matrix can be used to rerank initial retrieval results, and high-confidence pseudo-labels can be generated by the refined distance matrix for providing substantial and diverse supervisory signals. Despite the importance of neighborhood refinery, it remains a challenging task to efficiently identify the robust neighbors within the data manifold.

Recently, several innovative methods have been developed to enhance the robustness of neighborhood refinery, notably through diffusion-based and context-based approaches. To reveal the intrinsic relationship within the underlying data manifold, diffusion-based methods (Iscen et al., 2017; 2018; Bai et al., 2019; Zhang et al., 2023; Yang et al., 2019) perform similarity propagation within the $k$-nearest neighbor graph. However, the negative impact of outliers inevitably propagates throughout the graph, leading to a decrease in the discrimination of the resulting similarity matrix used for neighborhood refinery. In addition, context-based methods (Kim et al., 2022; Yu et al., 2023; Chen et al., 2024; Zhong et al., 2017; Zhang et al., 2020) are proposed to map the original feature into a contextual affinity feature space by encoding each instance with its local neighbors. Since consistent

neighbors are supposed to share high similarity within the contextual affinity space, they are effective in adjusting the similarity matrix. Despite their effectiveness, these methods do not adequately capture global relationships, making the feature representation susceptible to neighborhood outliers. Thus, it is imperative to improve global awareness of the manifold and reduce the impact of outliers to improve the performance of the neighborhood refinery.

Therefore, a reasonable approach is to construct an affinity graph from the original Euclidean space to intrinsically capture the data manifold information, in which the global-aware distribution can be formulated under the constraint of edge weights in a diffusion manner. To comprehensively perceive the manifold structure, the pairwise distances can be measured by transporting the global-aware distributions along the affinity graph. Moreover, considering the high consistency within the local neighbors, the influence of outliers can be further mitigated by integrating the neighbor distributions into a barycenter. Unlike trivial solutions that simply average neighboring distributions, the Euclidean distance from the barycenter to local neighbors can be jointly minimized with the Wasserstein distance (Solomon et al., 2015; Lin et al., 2020; Cuturi & Doucet, 2014) to enhance the perception of the neighborhood structure. After obtaining the robust barycenter distributions, we incorporate global geometric spatial information to distinguish instances that are not in the close neighborhood domain more effectively. This can be achieved by transporting (Cuturi, 2013; Janati et al., 2020) the global-aware distributions along the geodesic path within the affinity graph, where the resulting distance can be used for neighborhood refinery.

In this work, we propose a novel Neighbor-aware Geodesic Transportation (NGT) to exploit the underlying manifold information for neighborhood refinery, consisting of Manifold-aware Neighbor Encoding (MNE), Regularized Barycenter Refinery (RBR) and Geodesic Transportation (GT). First, by taking advantage of the diffusion process, Manifold-aware Neighbor Encoding (MNE) embeds each instance into a manifold-aware feature space. The resulting distributions are regularized by the affinity weights of a $k$-nearest neighbor graph constructed from the original Euclidean space, allowing the global manifold relationships between each instance and the entire dataset to be captured. Afterward, a Regularized Barycenter Refinery (RBR) is proposed to integrate local neighbors into a barycenter to mitigate the negative effects of outliers. Formally, the Euclidean distance from barycenter to local neighbors and the Wasserstein distance serving as a regularization term is jointly minimized. This approach implicitly considers the pairwise connections between neighborhood instances, enabling us to obtain more robust and global-aware distributions. Finally, we introduce Geodesic Transportation (GT) to measure the pairwise distances among distributions by formulating it as an optimal transport problem. Specifically, distributions are transported along geodesic paths within the affinity graph, inherently incorporating the geometric and global context information of the data manifold. To maintain consistency in Euclidean space, we combine the resulting transportation distance with the original Euclidean distance to perform neighborhood refinery.

The proposed Neighbor-aware Geodesic Transportation (NGT) can be easily adapted to the image retrieval tasks for re-ranking and the self-supervised framework for deep clustering. Extensive experiments have shown the superiority of our proposed method. Specifically, NGT achieves the mAP of 81.1%/91.7% on $R$Oxf(M) and $R$Par(M) respectively. Moreover, after conducting NGT to deep clustering tasks, the obtained performance surpasses the top-performs model by 2.4%/1.1%/2.1% in NMI/ACC/ARI on CIFAR-20 respectively.

## 2 RELATED WORK

Neighborhood refinery tackles the challenge of identifying semantically similar neighbors for each sample by considering the distance relationships among all samples in the feature space. This process, known as re-ranking in the context of image retrieval, serves as a post-processing approach that can significantly enhance retrieval performance. Similarly, it can also provide richer supervisory signals for training in deep clustering, improving both feature learning and clustering results.

### 2.1 RE-RANKING

Image retrieval seeks to identify images with similar content from a large database, while re-ranking is a training-free technique that improves the overall retrieval performance by refining the initial ranking list through a second retrieval or optimization process. Existing re-ranking methods can

be broadly classified into four categories: *Query Expansion*, *learning-based*, *context-based*, and *diffusion-based* methods. Query expansion methods (Chum et al., 2007; Gordo et al., 2017; Radenović et al., 2019; Shao et al., 2023) focus on integrating neighbor image features through various averaging and weighting strategies to build a more effective query. Meanwhile, learning-based methods involve approaches utilizing self-attention mechanisms (Ouyang et al., 2021; Gordo et al., 2020) for adjustable weight aggregation, along with techniques that leverage graph neural networks (Liu et al., 2019; Shen et al., 2021a) to enhance sample perception across the entire feature space.

The insight that the contextual information captured by $k$-nearest neighbors can enhance retrieval performance has prompted the development of context-based re-ranking. Shen et al. (2012); Sarfraz et al. (2018) update the distance measure using the rank lists of $k$-nearest neighbors. Additionally, Yu et al. (2023); Chen et al. (2024); Kim et al. (2022); Zhang et al. (2020); Zhong et al. (2017) encode each instance into a contextual affinity space to perform neighborhood refinery, ensuring similar images have higher consistency. Leveraging the intrinsic manifold structure of data, diffusion-based methods serve as powerful neighborhood refinery techniques. Prominent works (Iscen et al., 2018; Yang et al., 2019; Luo et al., 2024; Zhang et al., 2023) employ the affinity graph to represent the underlying data manifold and have shown remarkable performance in image retrieval, while Bai et al. (2019); Chang et al. (2019); Zhou et al. (2012) further incorporate the hypergraphs to effectively aggregate higher-order information, leading to better performance.

## 2.2 DEEP CLUSTERING AND SELF-SUPERVISED LEARNING

Deep clustering seeks to simultaneously learn image representations and conduct clustering in an integrated manner, while neighborhood refinery can serve as an adaptive module to provide richer information during the training stage. Over the past few decades, substantial research endeavors (Metaxas et al., 2023; Cai et al., 2023; Shiran & Weinshall, 2021; Li et al., 2022; 2024) have been directed towards this task. Recently, driven by self-supervised learning, deep clustering has advanced significantly. For instance, IDFD (Tao et al., 2021), MoCo (He et al., 2020), SimCLR (Chen et al., 2020) and ProPos (Huang et al., 2023) leverage pretext tasks and construct contrastive losses for representation learning, while approaches such as BYOL (Grill et al., 2020) and SwAV Caron et al. (2020) introduce a non-contrastive paradigm. The incorporation of self-supervised methods has enriched the learned features and enhanced clustering performance. Building on these approaches, the supervisory information embedded in neighboring samples has also been utilized to guide feature learning. Dwibedi et al. (2021); Navaneet et al. (2022); Yu et al. (2023) demonstrate that leveraging semantically similar neighbor instances can further enhance the robustness and quality of learned features. Thus, employing neighborhood refinery techniques to effectively identify and aggregate neighbor information represents an important direction for deep clustering.

## 3 METHODOLOGY

The Neighborhood Refinery aims to adjust for better neighborhood relationships by utilizing information from the underlying structure of the data manifold. We propose a novel Neighbor-aware Geodesic Transportation (NGT) (Figure 1). comprising Manifold-aware Neighbor Encoding (MNE), Regularized Barycenter Refinery (RBR), and Geodesic Transportation (GT) to capture both global relationships and local features. Global relationships refer to interactions across the entire dataset, while local features reflect immediate connections between closely situated entities. MNE first embeds each instance into a manifold-aware space under the supervision of an affinity graph constructed from Euclidean space to enhance the perception of the global manifold structure. After that, to mitigate the negative effect of outliers, RBR integrates local neighbors into a robust barycenter distribution by adding a Wasserstein regularization term to constrain the pairwise relationships among neighbors. Finally, GT transports the distributions along the geodesic path within the affinity graph, combining geometric and global context information to generate the refined distance matrix.

### 3.1 MANIFOLD-AWARE NEIGHBOR ENCODING

To address the problem that existing context-based methods cannot capture the global relationships, Manifold-aware Neighbor Encoding (MNE) aims to project each instance into a manifold-aware space, in which the corresponding distributions are under the constraint of the original affinity graph

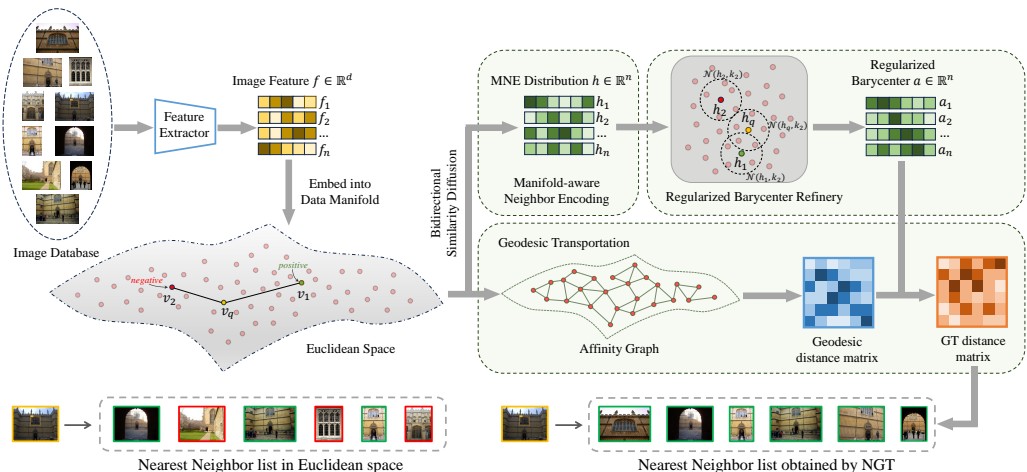

Figure 1: The framework of Neighbor-aware Geodesic Transportation (NGT). Relative distance relationships in Euclidean space lack discriminative power, *e.g.*, $d(v_q, v_2) < d(v_q, v_1)$ (expected to be $d(v_q, v_2) > d(v_q, v_1)$). The proposed NGT first encodes features into a manifold-aware space and then uses Regularized Barycenter Refinery (RBR) to integrate local neighbors into a robust barycenter distribution. Geodesic Transportation (GT) calculates pairwise distances by propagating distributions on the nearest neighbor graph, resulting in improved neighbor relationships.

constructed from Euclidean space. Formally, given a batch set $\mathcal{X} = \{x_1, x_2, \ldots, x_n\}$ comprising $n$ images, we can obtain the corresponding $d$-dimensional features $\mathcal{F} = \{\boldsymbol{f}_1, \boldsymbol{f}_2, \ldots, \boldsymbol{f}_n\}$. The pairwise Euclidean distance $d(i, j)$ between images $x_i$ and $x_j$ can be formulated as:

$$d(i, j) = \|\boldsymbol{f}_i - \boldsymbol{f}_j\|_2. \tag{1}$$

For an image $x_i$, its $k$-nearest neighbors within the Euclidean space are denoted as $\mathcal{N}(i, k)$. The underlying manifold structure can be represented by an affinity graph $\mathcal{G} = \{\mathcal{V}, \mathcal{E}\}$ constructed by the whole dataset, where each element $v_i$ in vertices set $\mathcal{V}$ corresponds to an image in $\mathcal{X}$, and the set of edges $\mathcal{E}$ denotes the connections between pair of vertices. Therefore, the affinity weights $\boldsymbol{W}_{ij}$ generated with the Gaussian kernel function is:

$$\boldsymbol{W}_{ij} = \mathbb{1}_{ij}^{\mathcal{N}} \exp\left(-d^2(i, j)/\sigma^2\right), \tag{2}$$

where $\mathbb{1}^{\mathcal{N}}$ is an indicator matrix with its element $\mathbb{1}_{ij}^{\mathcal{N}} = 1$ if $j \in \mathcal{N}(i, k)$, represents that the affinity graph is sparse where $k$-nearest neighbors are connected. And $\sigma$ is the hyper-parameter.

Compared to the Euclidean space, the manifold-aware space can not only enhance the incorporation of contextual information but also improve the overall perception of the underlying manifold. Formally, the corresponding sparse features in the manifold-aware space are denoted as $\mathcal{H} = \{\boldsymbol{h}_1, \boldsymbol{h}_2, \ldots, \boldsymbol{h}_n\} \in \mathbb{R}^n$. To determine which elements are suitable for representing the sparse feature $\boldsymbol{h}_i$, we introduce the $k$-reciprocal (Qin et al., 2011; Zhong et al., 2017) strategy to identify a set of related images $\mathcal{R}(i, k)$ that are mutually among the top-$k$ nearest neighbors. This approach ensures robust relevance among neighbors, and the formal definition is as follows:

$$\mathcal{R}(i, k) = \{j | (j \in \mathcal{N}(i, k)) \wedge (i \in \mathcal{N}(j, k))\}. \tag{3}$$

We set the value of $k$ used for manifold aware feature encoding as $k_1$ to avoid ambiguity, thus for an image $x_i$, the majority of the elements in $\boldsymbol{h}_i$ are zero, except for those indices that belong to the reciprocal set $\mathcal{R}(i, k_1)$, where $\mathcal{R}(i, k_1) \subseteq \{1, 2, \ldots, n\}$ and $|\mathcal{R}(i, k_1)| < n$.

To address the problem that existing context-based methods lack the ability to reveal the intrinsic relationship in the underlying data manifold, we incorporate the diffusion process (Bai et al., 2019; Luo et al., 2024) to obtain a manifold-aware similarity matrix $\boldsymbol{F}$ for encoding the sparse feature set $\mathcal{H}$. Specifically, here we adopt the Bidirectional Similarity Diffusion strategy, followed by:

$$\min_{\boldsymbol{F}} \frac{1}{4} \sum_{k=1}^{n} \sum_{i,j=1}^{n} \left( \boldsymbol{W}_{ij} \left( \frac{\boldsymbol{F}_{ki}}{\sqrt{\boldsymbol{D}_{ii}}} - \frac{\boldsymbol{F}_{kj}}{\sqrt{\boldsymbol{D}_{jj}}} \right)^2 + \boldsymbol{W}_{ij} \left( \frac{\boldsymbol{F}_{ik}}{\sqrt{\boldsymbol{D}_{ii}}} - \frac{\boldsymbol{F}_{jk}}{\sqrt{\boldsymbol{D}_{jj}}} \right)^2 \right) + \mu \|\boldsymbol{F} - \boldsymbol{E}\|_F^2, \tag{4}$$

where $\boldsymbol{D}$ is a diagonal matrix with its $i$-th diagonal element equal to the summation of the $i$-th row in $\boldsymbol{W}$. The semidefinite matrix $\boldsymbol{E}$ in the regularization term is used to prevent the diffusion similarity $\boldsymbol{F}$ from being excessively smooth. The choice of $\boldsymbol{E}$ can be a diagonal matrix $\boldsymbol{I}$ or approximated by the affinity relationship such as $(\boldsymbol{W} + \boldsymbol{W}^\top)/2$, and the hyper-parameter $\mu > 0$ is a constraint weight.

As demonstrated in Section A, directly solving the closed-form solution to Equation 4 is computationally expensive. By employing the iterative method described below, the time complexity can be reduced to $\mathcal{O}(n^3)$,

$$\boldsymbol{F}^{(t+1)} = \frac{1}{2}\alpha \boldsymbol{F}^{(t)}\bar{\boldsymbol{S}}^\top + \frac{1}{2}\alpha\bar{\boldsymbol{S}}\boldsymbol{F}^{(t)} + (1-\alpha)\boldsymbol{E}, \tag{5}$$

where $\alpha = \frac{1}{1+\mu}$ and $\bar{\boldsymbol{S}} = (\boldsymbol{S} + \boldsymbol{S}^\top)/2$. Through conjugate gradient descent, it can be approximated more efficiently with fewer iteration steps following Algorithm 1. After the optimal similarity matrix $\boldsymbol{F}$ is obtained, we can encode the manifold-aware distribution for each images $x_i$ as $\boldsymbol{h}_i = \mathbb{1}_i^\mathcal{R}\boldsymbol{F}_i$, where $\mathbb{1}^\mathcal{R}$ is an indicator for searching reciprocal neighbors, $i.e.$, $\mathbb{1}_{ij}^\mathcal{R} = 1$, if $j \in \mathcal{R}(i, k_1)$. Therefore, we can transform all original image features $\mathcal{F} = \{\boldsymbol{f}_1, \boldsymbol{f}_2, \ldots, \boldsymbol{f}_n\}$ into manifold-aware distributions $\mathcal{H} = \{\boldsymbol{h}_1, \boldsymbol{h}_2, \ldots, \boldsymbol{h}_n\}$, this procedure is named as Manifold-aware Neighbor Encoding.

## 3.2 REGULARIZED BARYCENTER REFINERY

Despite the global-aware distributions being beneficial in capturing the global underlying manifold, they still suffer from the problem that the expressiveness is affected by outliers from the neighbor domain. Since local neighbors exhibit higher consistency, we can integrate neighbor distributions into a barycenter to enhance the representativeness and mitigate the negative influence of outliers. Unlike trivial solutions that simply average neighboring distributions together, we aim to consider pairwise connections among neighborhood instances. To achieve this, we introduce a Wasserstein distance serving as a regularization term to jointly minimize the Euclidean distance from the barycenter to the local neighbors.

We therefore give a more general formulation for integrating $n$ distributions into a refined barycenter $\boldsymbol{a}$ weighted by a set of weights $\lambda = \{\lambda_1, \lambda_2, \ldots, \lambda_n\}$. Without losing of generality, the sum of $\lambda$ is explicitly constrained to 1, $i.e.$, $\sum_s \lambda_s = 1$. For the case where only the local neighbors of a given image $x_i$ need to be integrated, we can define the indices of its local neighbors as $\mathcal{N}(i, k_2)$, where $k_2 < k_1$, and explicitly set $\lambda_s = 0$ for $s \notin \mathcal{N}(i, k_2)$. The weighted summation of Euclidean distance and Wasserstein distance is jointly minimized with the following optimization problem,

$$\min_{\boldsymbol{P}, \boldsymbol{a}_i} \quad \sum_{s=1}^n (1-\omega)\lambda_s\langle\boldsymbol{a}_i - \boldsymbol{h}_s, \boldsymbol{a}_i - \boldsymbol{h}_s\rangle + \omega\lambda_s\langle\boldsymbol{C}, \boldsymbol{P}_s\rangle),$$
$$\text{s.t.} \quad \boldsymbol{P}_s\mathbf{1} = \boldsymbol{a}_i, \boldsymbol{P}_s^\top\mathbf{1} = \boldsymbol{h}_s, \tag{6}$$

where $\boldsymbol{P}_s$ is a matrix that represents the optimal transport procedure from barycenter $\boldsymbol{a}$ to $\boldsymbol{h}_s$, all the $\boldsymbol{P}_s$ come together to make up the objective strategies $\boldsymbol{P}$, and $\langle\cdot\rangle$ is an inner product operator that denotes the summation of all the product between corresponding components. $\omega$ is the weight used to balance the contribution of Euclidean distance and Wasserstein distance in computing the barycenter.

In the minimization objective of Equation 6, the first term is the Euclidean distance, and the second term is the Wasserstein distance induced by the cost matrix $\boldsymbol{C}$. Formally, the cost matrix $\boldsymbol{C}$ is used to capture the pairwise relationships between neighboring instances, which can be measured by the similarity within the Euclidean space or treated uniformly with a 0-1 matrix, $i.e.$, $\boldsymbol{C}_{ij} = 1$ when $i = j$, and $\boldsymbol{C}_{ij} = 0$ when $i \neq j$, to ensure more stable performance in various scenarios. Following Agueh & Carlier (2011); Cuturi (2013); Cuturi & Doucet (2014); Solomon et al. (2015), an entropy regularization term $H(\boldsymbol{P}_s) = \sum_{ij} \boldsymbol{P}_{ij}^s - \boldsymbol{P}_{ij}^s \log \boldsymbol{P}_{ij}^s$ is added to the Wasserstein distance term in order to achieve better numerical stability and enable the efficient iterative solution. The entropy regularized problem is formulated as follows:

$$\min_{\boldsymbol{P}, \boldsymbol{a}_i} \quad \sum_{s=1}^n (1-\omega)\lambda_s\langle\boldsymbol{a}_i - \boldsymbol{h}_s, \boldsymbol{a}_i - \boldsymbol{h}_s\rangle + \omega\lambda_s\big(\langle\boldsymbol{C}, \boldsymbol{P}_s\rangle - \varepsilon H(\boldsymbol{P}_s)\big),$$
$$\text{s.t.} \quad \boldsymbol{P}_s\mathbf{1} = \boldsymbol{a}_i, \boldsymbol{P}_s^\top\mathbf{1} = \boldsymbol{h}_s. \tag{7}$$

To solve the minimization problem in Equation 7, we define the Lagrange multipliers for the two equality constraints as $\boldsymbol{g} = \{\boldsymbol{g}_1, \boldsymbol{g}_2, \ldots, \boldsymbol{g}_n\}$ and $\boldsymbol{r} = \{\boldsymbol{r}_1, \boldsymbol{r}_2, \ldots, \boldsymbol{r}_n\}$ respectively. After that, the

Lagrangian function $\mathcal{L}(\boldsymbol{P}, \boldsymbol{a}, \boldsymbol{g}, \boldsymbol{r})$ of the primal entropy regularized minimization problem can be formulated as:

$$
\begin{aligned}
\mathcal{L}(\boldsymbol{P}, \boldsymbol{a}, \boldsymbol{g}, \boldsymbol{r}) = \sum_{s=1}^{n} &(1-\omega)\lambda_s \langle \boldsymbol{a} - \boldsymbol{h}_s, \boldsymbol{a} - \boldsymbol{h}_s \rangle + \lambda_s \omega \big( \langle \boldsymbol{C}, \boldsymbol{P}_s \rangle - \varepsilon H(\boldsymbol{P}_s) \big) \\
&+ \lambda_s \omega \big( \langle \boldsymbol{g}_s, \boldsymbol{a} - \boldsymbol{P}_s \mathbf{1} \rangle + \langle \boldsymbol{r}_s, \boldsymbol{h}_s - \boldsymbol{P}_s^\top \mathbf{1} \rangle \big).
\end{aligned}
\tag{8}
$$

The objective function is strictly convex and such that the strong duality holds. Therefore, solving the primal problem is equivalent to find the maximum value of the following dual Lagrangian function,

$$
\mathcal{D}(\boldsymbol{g}, \boldsymbol{r}) = \inf_{\boldsymbol{P}, \boldsymbol{a}} \mathcal{L}(\boldsymbol{P}, \boldsymbol{a}, \boldsymbol{g}, \boldsymbol{r}).
\tag{9}
$$

By solving the dual problem of $\max_{\boldsymbol{g}, \boldsymbol{r}} \mathcal{D}(\boldsymbol{g}, \boldsymbol{r})$, we can derive the representation of the optimal solution of each transportation matrix $\boldsymbol{P}_s^*$ as:

$$
\boldsymbol{P}_s^* = \operatorname{diag}(e^{\boldsymbol{g}_s/\varepsilon}) \boldsymbol{K} \operatorname{diag}(e^{\boldsymbol{r}_s/\varepsilon}),
\tag{10}
$$

and the following relationship can also be established:

$$
\operatorname{diag}(e^{\boldsymbol{g}_s/\varepsilon}) \boldsymbol{K} e^{\boldsymbol{r}_s/\varepsilon} + \xi \sum_{s=1}^{n} \lambda_s \boldsymbol{g}_s = \sum_{s=1}^{n} \lambda_s \boldsymbol{h}_s,
\tag{11}
$$

where $\boldsymbol{K}$ is a variant matrix of $\boldsymbol{C}$ with its element defined as $\boldsymbol{K}_{ij} = e^{-\boldsymbol{C}_{ij}/\varepsilon}$, and operator $\operatorname{diag}(\cdot)$ can transform vector into a corresponding diagonal matrix. Here we substitute $\xi = \frac{\omega}{2(1-\omega)}$ for the case of simplicity. As the primal problem is strictly convex, the optimal value can be approximated following the fixed point theory. The algorithm and derivation procedures are detailed in Appendix B. To further enhance the numerical stability, we can estimate the optimal solution by solving the optimization problem that involves the Euclidean distance term and the Wasserstein distance term separately. The algorithm and derivations are discussed in Appendix C. The resulting barycenter distributions for each instance are denoted as $\{\boldsymbol{a}_1, \boldsymbol{a}_2, \ldots, \boldsymbol{a}_n\}$.

### 3.3 GEODESIC TRANSPORTATION

After that, the next goal is to devise an efficient distance metric among barycenter distributions that is capable of integrating global contextual information. Conventional methods often fail to adequately discern instances outside the immediate neighborhood domain and are unable to fully exploit geometric information. To overcome these limitations, we focus on transporting distributions along geodesic paths to compute pairwise distances for neighborhood refinery, which incorporate geodesic distances into the computation of Wasserstein distance.

First we define the set $\mathcal{P}_{s \to t}$, which contains all the possible path from vertex $v_s$ to $v_t$ within the affinity graph $\mathcal{G} = \{\mathcal{V}, \mathcal{E}\}$:

$$
\mathcal{P}_{s \to t} = \{(u_1, u_2, \ldots, u_k) | u_1 = v_s, u_k = v_t, \boldsymbol{e}[u_i, u_{i+1}] \in \mathcal{E}, \forall\, 1 \le i < k \},
\tag{12}
$$

where each path $p$ in $\mathcal{P}_{s \to t}$ is a sequence of of vertex that starting from $v_s$ and end with $v_t$, $u_1$ to $u_k$ correspond to vertex in the affinity graph. $\boldsymbol{e}[u_i, u_{i+1}]$ denotes the edge between the vertex $u_i$ and $u_{i+1}$. The geodesic path from $v_s$ to $v_t$ is also equivalent to the shortest path in the affinity graph, which can incorporate the topological structure of the data manifold and capture global information. Moreover, the geodesic distance $\boldsymbol{C}'_{st}$ from $v_s$ to $v_t$ is defined as:

$$
\boldsymbol{C}'_{st} = \min_{p \in \mathcal{P}_{s \to t}} \sum_{e_{ij} \in p} \boldsymbol{W}_{ij},
\tag{13}
$$

which can be viewed as the summation of the edge weights along the geodesic path, and all pairs of shortest distance formulate the geodesic distance matrix $\boldsymbol{C}'$. This distance matrix will serve as the cost matrix in the subsequent discussion. To emphasize its physical significance and maintain clarity, we adopt the same notation as in the previous section. Besides, the problem of searching for the geodesic path can be solved efficiently by taking the advantage from Saha & Ye (2024).

To integrate the global correlation capability of optimal transport with the geometric properties of geodesic distances, we leverage the geodesic path into the transporting process and use it as the

cost matrix for Wasserstein distance. Given two barycenter distributions $\boldsymbol{a}_i$ and $\boldsymbol{a}_j$, to measure the pairwise distance, we need to solve:

$$\min_{\boldsymbol{P}} \quad \langle \boldsymbol{C}', \boldsymbol{P} \rangle - \varepsilon H(\boldsymbol{P})$$
$$\text{s.t.} \quad \boldsymbol{P}\mathbf{1} = \boldsymbol{a}_i, \boldsymbol{P}^\top \mathbf{1} = \boldsymbol{a}_j, \tag{14}$$

where matrix $\boldsymbol{P}$ is the optimal transportation matrix from $\boldsymbol{a}_i$ to $\boldsymbol{a}_j$. To balance the benefits of optimal transport distance with computational efficiency, here we also incorporate $H(\boldsymbol{P}) = \sum_{ij} \boldsymbol{P}_{ij} - \boldsymbol{P}_{ij} \log \boldsymbol{P}_{ij}$ as an entropy regularization term. Such that the solution to the optimal transport problem can be numerically solved following Sinkhorn-Knopp iterations (Cuturi, 2013; Peyré & Cuturi, 2020). After obtaining the optimal transportation matrix $\boldsymbol{P}$ from $\boldsymbol{a}_i$ to $\boldsymbol{a}_j$, the distance between two distributions can be formulated as:

$$d'(i, j) = \langle \boldsymbol{P}, \boldsymbol{C}' \rangle = \sum_{k,l=1}^{n} \boldsymbol{P}_{kl} \boldsymbol{C}'_{kl} \tag{15}$$

Furthermore, to preserve the crucial neighborhood information within the original Euclidean space, we simultaneously incorporate the original Euclidean distance and the optimal transport-based distance. The final Geodesic Transportation distance between two distributions can be formulated as:

$$d_{GT}(i, j) = (1 - \theta)d'(i, j) + \theta d(i, j), \tag{16}$$

where $d(i, j)$ is the Euclidean distance, and $\theta$ is the balance weight. All pairwise distances among barycenter distributions can be calculated in a parallel manner, resulting in a refined distance matrix that can be used for neighborhood refinery.

## 4 EXPERIMENT

### 4.1 EXPERIMENT SETUP

**Datasets:** To verify the effectiveness of our proposed NGT, we conduct experiments on both image retrieval and deep clustering tasks. For image retrieval, the well-known Oxford5k (Philbin et al., 2007) and Paris6k (Philbin et al., 2008) datasets have been revisited by Radenović et al. (2018), referred to as *R*Oxf and *R*Par respectively. Additionally, the performance on large-scale datasets with an extra 1 million distractor images, named *R*Oxf+1M and *R*Par+1M, has also been evaluated. For deep clustering tasks, the performance is measured on five widely used benchmarks, including CIFAR-10 (Krizhevsky et al., 2009), CIFAR-20 (Krizhevsky et al., 2009), STL-10 (Coates et al., 2011), ImageNet-10 (Chang et al., 2017), and ImageNet-Dogs (Chang et al., 2017).

**Evaluation Metrics:** For image retrieval tasks, the image datasets are categorized into three levels of difficulty by Radenović et al. (2018), and we employ mean Average Precision (mAP) to measure the retrieval performance. For deep clustering tasks, we strictly follow Huang et al. (2023); Shen et al. (2021b); Yu et al. (2023); Li et al. (2021; 2022) to adopt Normalized Mutual Information (NMI), Accuracy (ACC), and Adjusted Rand Index (ARI) for evaluation.

**Implementations:** The image descriptors for instance retrieval tasks are extracted by MAC (Tolias et al., 2016), R-MAC (Tolias et al., 2016), R-GeM (Radenović et al., 2019), DOLG (Yang et al., 2021), and CVNet (Lee et al., 2022), respectively. For the deep clustering task, we apply our method to the BYOL (Grill et al., 2020) framework as a post-training stage. Following Li et al. (2021); Tsai et al. (2020); Huang et al. (2023); Yu et al. (2023), the training details, including the choice of batch size, hyper-parameters, and backbone architecture, are summarized in Appendix D.

### 4.2 COMPARISON WITH EXISTING METHODS

**Comparison of Image Retrieval:** As summarized in Table 1, we evaluate the retrieval performance with existing re-ranking method, including query expansion methods (AQE (Chum et al., 2007), $\alpha$QE (Radenović et al., 2019), DQE (Arandjelović & Zisserman, 2012), AQEwD (Gordo et al., 2017) and SG (Shao et al., 2023)), context-based methods (STML (Kim et al., 2022), ConAff (Yu et al., 2023) and CAJ (Chen et al., 2024)), diffusion-based methods (DFS (Iscen et al., 2017), FSR (Iscen et al., 2018), RDP (Bai et al., 2019), CAS(Luo et al., 2024), GSS (Liu et al., 2019) and EGT (Chang et al.,

Table 1: Evaluation of the retrieval performances based on R-GeM (Radenović et al., 2019).

| Method | Medium | | | | Hard | | | |
|---|---|---|---|---|---|---|---|---|
| | $R$Oxf | $R$Oxf+1M | $R$Par | $R$Par+1M | $R$Oxf | $R$Oxf+1M | $R$Par | $R$Par+1M |
| R-GeM (Radenović et al., 2019) | 67.3 | 49.5 | 80.6 | 57.4 | 44.2 | 25.7 | 61.5 | 29.8 |
| AQE (Chum et al., 2007) | 72.3 | 56.7 | 82.7 | 61.7 | 48.9 | 30.0 | 65.0 | 35.9 |
| $\alpha$QE (Radenović et al., 2019) | 69.7 | 53.1 | 86.5 | 65.3 | 44.8 | 26.5 | 71.0 | 40.2 |
| DQE (Arandjelović & Zisserman, 2012) | 70.3 | 56.7 | 85.9 | 66.9 | 45.9 | 30.8 | 69.9 | 43.2 |
| AQEwD (Gordo et al., 2017) | 72.2 | 56.6 | 83.2 | 62.5 | 48.8 | 29.8 | 65.8 | 36.6 |
| LAttQE (Gordo et al., 2020) | 73.4 | 58.3 | 86.3 | 67.3 | 49.6 | 31.0 | 70.6 | 42.4 |
| ADBA+AQE | 72.9 | 52.4 | 84.3 | 59.6 | 53.5 | 25.9 | 68.1 | 30.4 |
| $\alpha$DBA+$\alpha$QE | 71.2 | 55.1 | 87.5 | 68.4 | 50.4 | 31.7 | 73.7 | 45.9 |
| DDBA+DQE | 69.2 | 52.6 | 85.4 | 66.6 | 50.2 | 29.2 | 70.1 | 42.4 |
| ADBAwD+AQEwD | 74.1 | 56.2 | 84.5 | 61.5 | 54.5 | 31.1 | 68.6 | 33.7 |
| LAttDBA+LAttQE | 74.0 | 60.0 | 87.8 | 70.5 | 54.1 | 36.3 | 74.1 | 48.3 |
| DFS (Iscen et al., 2017) | 72.9 | 59.4 | 89.7 | 74.0 | 50.1 | 34.9 | 80.4 | 56.9 |
| FSR (Iscen et al., 2018) | 72.7 | 59.6 | 89.6 | 73.9 | 49.6 | 34.8 | 80.2 | 56.7 |
| RDP (Bai et al., 2019) | 75.2 | 55.0 | 89.7 | 70.0 | 58.8 | 33.9 | 77.9 | 48.0 |
| GSS (Liu et al., 2019) | 78.0 | 61.5 | 88.9 | 71.8 | 60.9 | 38.4 | 76.5 | 50.1 |
| EGT (Chang et al., 2019) | 74.7 | 60.1 | 87.9 | 72.6 | 51.1 | 36.2 | 76.6 | 51.3 |
| SG (Shao et al., 2023) | 71.4 | 53.9 | 83.6 | 61.5 | 49.5 | 28.8 | 67.6 | 35.8 |
| SSR (Shen et al., 2021a) | 74.2 | 54.6 | 82.5 | 60.0 | 53.2 | 29.3 | 65.6 | 35.0 |
| CSA (Ouyang et al., 2021) | 78.2 | 61.5 | 88.2 | 71.6 | 59.1 | 38.2 | 75.3 | 51.0 |
| STML (Kim et al., 2022) | 74.1 | 53.5 | 85.4 | 68.0 | 57.1 | 27.5 | 70.0 | 42.9 |
| ConAff (Yu et al., 2023) | 74.5 | 53.9 | 88.0 | 61.4 | 56.4 | 30.3 | 73.9 | 33.6 |
| **NGT (Ours)** | **81.1** | **61.6** | **91.7** | **75.8** | **64.5** | **39.0** | **81.5** | **58.8** |

Table 2: Evaluation of the retrieval performances based on DOLG (Yang et al., 2021), best in **bold**.

| Method | Easy | | Medium | | Hard | |
|---|---|---|---|---|---|---|
| | $R$Oxf | $R$Par | $R$Oxf | $R$Par | $R$Oxf | $R$Par |
| DOLG | 93.4 | 95.2 | 81.2 | 90.1 | 62.6 | 79.2 |
| AQE | 96.0 | 95.6 | 83.5 | 90.5 | 67.5 | 80.0 |
| $\alpha$QE | 96.7 | 95.7 | 83.9 | 91.4 | 67.6 | 81.7 |
| SG | 97.7 | 95.7 | 85.1 | 91.7 | 70.3 | 82.9 |
| CAJ | 96.0 | 94.2 | 85.8 | 91.2 | 71.8 | 81.3 |
| STML | 97.6 | 95.4 | 86.0 | 91.5 | 70.8 | 82.3 |
| DFS | 87.3 | 93.6 | 76.1 | 90.8 | 53.5 | 82.4 |
| RDP | 95.7 | 95.0 | 87.2 | 93.0 | 72.0 | 84.8 |
| CAS | 96.8 | 95.7 | 89.5 | 93.6 | **76.7** | 86.7 |
| GSS | 98.0 | 95.3 | 86.9 | 90.6 | 72.9 | 81.2 |
| ConAff | 95.1 | 93.0 | 84.6 | 91.3 | 66.7 | 79.9 |
| **NGT** | **99.1** | **96.1** | **90.3** | **95.0** | 76.5 | **89.2** |

Table 3: Evaluation of the retrieval performances based on CVNet (Lee et al., 2022), best in **bold**.

| Method | Easy | | Medium | | Hard | |
|---|---|---|---|---|---|---|
| | $R$Oxf | $R$Par | $R$Oxf | $R$Par | $R$Oxf | $R$Par |
| CVNet | 94.3 | 93.9 | 81.0 | 88.8 | 62.1 | 76.5 |
| AQE | 94.7 | 94.4 | 82.1 | 90.2 | 64.4 | 78.8 |
| $\alpha$QE | 95.8 | 94.8 | 95.8 | 90.9 | 63.5 | 80.4 |
| SG | 99.0 | 95.9 | 86.1 | 90.6 | 69.3 | 80.5 |
| CAJ | 97.3 | 93.9 | 85.8 | 88.9 | 70.0 | 76.4 |
| STML | 98.5 | 94.9 | 86.2 | 90.8 | 69.3 | 80.5 |
| DFS | 83.5 | 93.5 | 70.8 | 89.8 | 47.4 | 79.6 |
| RDP | 96.9 | 94.5 | 87.8 | 92.4 | 71.5 | 83.3 |
| CAS | 97.6 | 95.0 | 87.6 | 92.8 | 72.7 | 84.8 |
| GSS | 99.0 | 94.0 | 87.6 | 87.1 | 70.4 | 76.9 |
| ConAff | 98.3 | 92.4 | 87.5 | 90.2 | 70.3 | 77.7 |
| **NGT** | **99.2** | **95.7** | **89.7** | **94.3** | **73.8** | **87.1** |

2019)), as well as learning-based methods (LAttQE (Gordo et al., 2020), SSR (Shen et al., 2021a) and CSA (Ouyang et al., 2021)). It can be observed that the proposed NGT achieves better performance than the existing methods in all settings. Among all existing methods, ConAff is the most related work to ours, which uses neighborhood for contextual encoding. Compared to ConAff, the proposed NGT obtains a 5.7%/9.8% improvement in mAP in $R$Oxf (M) and $R$Oxf (H) with the feature extracted by DOLG as shown in Table 2. The significant improvement in performance underscores the superiority of the proposed NGT, and more evaluated comparison results are provided in Appendix E.

**Comparison of Deep Clustering:** We further conduct a comparison with existing methods on the task of deep clustering in five datasets and summarize the related results in Table 4. Specifically, for the two moderate-sized datasets CIFAR-10 and CIFAR-20, the proposed NGT surpasses the baseline performance of BYOL by 7.8%/7.3% in NMI, respectively. Furthermore, compared to the best performing contrastive-based ProPos (Huang et al., 2023) and the non-contrastive-based ConNR (Yu et al., 2023), NGT obtains a 4.6% and 2.4% improvement upon them. Despite that the clustering performance on the smaller datasets STL-10 and ImageNet-10 nearly approaches saturation, our method still yields the optimal results, boasting an increasement of 14.2%/4.6% in NMI compared to the baseline BYOL. On the more challenging ImageNet-Dogs dataset, our method achieves state-of-the-art performance across two key metrics, indicating that there is still room for improvement. The results validate the effectiveness of NGT in enhancing the descriptive capabilities of self-supervised models for deep clustering.

Table 4: Deep clustering performance on five benchmarks.

| | CIFAR-10 | | | CIFAR-20 | | | STL-10 | | | ImageNet-10 | | | ImageNet-Dogs | | |
|---|---|---|---|---|---|---|---|---|---|---|---|---|---|---|---|
| | NMI | ACC | ARI | NMI | ACC | ARI | NMI | ACC | ARI | NMI | ACC | ARI | NMI | ACC | ARI |
| IIC (Ji et al., 2019) | 51.3 | 61.7 | 41.1 | - | 25.7 | - | 43.1 | 49.9 | 29.5 | - | - | - | - | - | - |
| DCCM (Wu et al., 2019) | 49.6 | 62.3 | 40.8 | 28.5 | 32.7 | 17.3 | 37.6 | 48.2 | 26.2 | 60.8 | 71.0 | 55.5 | 32.1 | 38.3 | 18.2 |
| PICA (Jiabo Huang & Zhu, 2020) | 56.1 | 64.5 | 46.7 | 29.6 | 32.2 | 15.9 | - | - | - | 78.2 | 85.0 | 73.3 | 33.6 | 32.4 | 17.9 |
| SCAN (Van Gansbeke et al., 2020) | 79.7 | 88.3 | 77.2 | 48.6 | 50.7 | 33.3 | 69.8 | 80.9 | 64.6 | - | - | - | - | - | - |
| NMM (Dang et al., 2021) | 74.8 | 84.3 | 70.9 | 48.4 | 47.7 | 31.6 | 69.4 | 80.8 | 65.0 | - | - | - | - | - | - |
| CC (Li et al., 2021) | 70.5 | 79.0 | 63.7 | 43.1 | 42.9 | 26.6 | 76.4 | 85.0 | 72.6 | 85.9 | 89.3 | 82.2 | 44.5 | 42.9 | 27.4 |
| MiCE (Tsai et al., 2020) | 73.7 | 83.5 | 69.8 | 43.6 | 44.0 | 28.0 | 63.5 | 75.2 | 57.5 | - | - | - | 42.3 | 43.9 | 28.6 |
| GCC (Zhong et al., 2021) | 76.4 | 85.6 | 72.8 | 47.2 | 47.2 | 30.5 | 68.4 | 78.8 | 63.1 | 84.2 | 90.1 | 82.2 | 49.0 | 52.6 | 36.2 |
| TCL (Li et al., 2022) | 81.9 | 88.7 | 78.0 | 52.9 | 53.1 | 35.7 | 79.9 | 86.8 | 75.7 | 87.5 | 89.5 | 83.7 | 62.3 | 64.4 | 51.6 |
| IDFD (Tao et al., 2021) | 71.1 | 81.5 | 66.3 | 42.6 | 42.5 | 26.4 | 64.3 | 75.6 | 57.5 | 89.8 | 95.4 | 90.1 | 54.6 | 59.1 | 41.3 |
| TCC (Shen et al., 2021b) | 79.0 | 90.6 | 73.3 | 47.9 | 49.1 | 31.2 | 73.2 | 81.4 | 68.9 | 84.8 | 89.7 | 82.5 | 55.4 | 59.5 | 41.7 |
| ProPos (Huang et al., 2023) | 85.1 | 91.6 | 83.5 | 58.2 | 57.8 | 42.3 | 75.8 | 86.7 | 73.7 | 89.6 | 95.6 | 90.6 | 73.7 | 77.5 | 67.5 |
| CoNR (Yu et al., 2023) | 86.7 | 93.2 | 86.1 | 60.4 | 60.4 | 44.3 | 85.2 | **92.6** | **84.6** | 91.1 | 96.4 | 92.2 | 74.4 | **79.4** | 66.7 |
| DivClust (Metaxas et al., 2023) | 72.4 | 81.9 | 68.1 | 44.0 | 43.7 | 28.3 | - | - | - | 89.1 | 93.6 | 87.8 | 51.6 | 52.9 | 37.6 |
| BYOL (Grill et al., 2020) | 79.4 | 87.8 | 76.6 | 55.5 | 53.9 | 37.6 | 71.3 | 82.5 | 65.7 | 86.6 | 93.9 | 87.2 | 63.5 | 69.4 | 54.8 |
| **NGT (Ours)** | **87.2** | **93.5** | **86.7** | **62.8** | **61.5** | **46.6** | **85.5** | 92.6 | 84.6 | **91.2** | **96.5** | **92.3** | **74.8** | 78.1 | **67.6** |

Table 5: Ablation Results on CIFAR-10 and CIFAR-20.

| BYOL | LN | MNE | RBR | GT | CIFAR-10 | | | CIFAR-20 | | |
|---|---|---|---|---|---|---|---|---|---|---|
| | | | | | NMI | ACC | ARI | NMI | ACC | ARI |
| ✓ | | | | | 79.4 | 87.8 | 76.6 | 55.5 | 53.9 | 37.6 |
| ✓ | ✓ | | | | 81.9 | 89.6 | 78.7 | 60.7 | 57.0 | 42.5 |
| ✓ | | ✓ | | | 83.5 | 89.5 | 80.1 | 61.7 | 58.3 | 43.7 |
| ✓ | | | ✓ | | 86.6 | 93.2 | 86.2 | 60.8 | 57.9 | 42.7 |
| ✓ | | | | ✓ | 86.7 | 93.3 | 86.3 | 61.3 | 58.4 | 43.6 |
| ✓ | | ✓ | | ✓ | 86.7 | 93.3 | 86.4 | 61.9 | 58.5 | 43.8 |
| ✓ | | ✓ | ✓ | | 86.7 | 93.3 | 86.3 | 62.5 | 59.8 | 45.2 |
| ✓ | | ✓ | ✓ | ✓ | **87.2** | **93.5** | **86.7** | **62.8** | **61.5** | **46.6** |

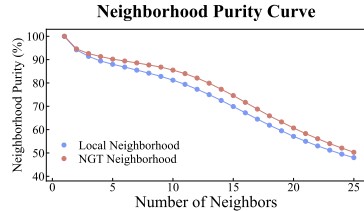

Figure 2: The neighborhood purity curve on CIFAR-10.

**Time Complexity Analysis:** Our method involves three main components: MNE, RBR, and GT. For MNE, we approximate the solution to the optimization problem in an iterative manner with Equation 5, achieving a time complexity of $\mathcal{O}(n^3)$. Similarly, RBR introduces entropy regularization term to find the fixed-point relationship for Equation 6, the iterative solution also results in a complexity of $\mathcal{O}(n^3)$. The GT module leverages the Sinkhorn-Knopp algorithm to handle the optimal transport problem, maintaining the same complexity. Consequently, the total time complexity of NGT is $\mathcal{O}(n^3)$. The comparisons of time complexity and re-ranking latency conducted on $R$Oxford are shown in Table 6, our method achieves a latency of 3,360 ms on GPU, demonstrating its outstanding performance. As for the large-scale datasets, we have implemented a coarse-to-fine re-ranking strategy, where our method is applied only to re-rank the top-$k$ images in the initial ranking list. Specifically, the latency remains within 4 seconds when $k$ equals 5,000, underscoring the effectiveness of our method.

## 4.3 ABLATION STUDY

**Effectiveness of each module:** We perform several ablation studies to evaluate the effectiveness of each module in deep clustering tasks, as shown in Table 5, where 'LN' indicates the Local Neighborhood obtained from Euclidean space. When applying MNE alone, the clustering results exceeded those of LN on both the CIFAR-10 and CIFAR-20 datasets, *e.g.*, outperforms LN by 1.6%/1.0% on CIFAR-10/20 datasets in terms of NMI. When RBR or GT is incorporated with a traditional context-based feature, a competitive result can be achieved, *e.g.,* RBR and GT improve the NMI/ACC/ARI from 83.5%/89.5%/80.1% to 86.6%/93.2%/86.2% and 86.7%/93.3%/86.3%, respectively, verifying the effectiveness of these two modules and their complementarity with MNE. Involving any of the two modules can provide different extents of enhancement, and the peak performance is achieved when all the modules are conducted to refine the neighborhood structure. Moreover, we present the neighbor purity curve during the training stage in Figure 2, the top-$k$ neighbors returned by our method exhibit higher accuracy compared to LN, further validating the effectiveness of NGT in identifying informative neighbors

**Ablations of MNE:** MNE encodes each instance into a sparse distribution, where the dimensions are determine by $k$-reciprocal neighbors and the values are assigned with diffusion-based similarity. As shown in Table 7, we also conduct the Cosine similarity in SCA (Bai & Bai, 2016) and the Gaussian similarity in Equation 2 to encode the sparse distributions. The diffusion-based similarity outperforms Cosine and Gaussian similarity by 5.7%/2.5% in mAP on $R$Oxf(H) based on R-GeM, demonstrate its

Table 6: Analysis of Time Complexity.

| Method | Time Complexity | Re-ranking Latency (ms) |
|--------|-----------------|--------------------------|
| $\alpha$QE | $\mathcal{O}(n^2)$ | 121 |
| DFS | $\mathcal{O}(n^3)$ | 2,129 |
| RDP | $\mathcal{O}(n^3)$ | 6,018 |
| STML | $\mathcal{O}(n^3)$ | 8,384 |
| GSS | - | > 5 min |
| NGT | $\mathcal{O}(n^3)$ | 3,360 |

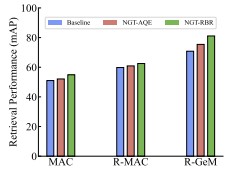
(a) Ablations of RBR on ROxf(M).

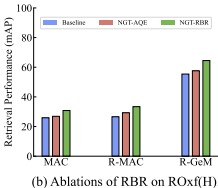
(b) Ablations of RBR on ROxf(H).

Figure 3: Ablations of RBR.

Table 7: Ablations of MNE.

| Method | ROxf(M) | | | ROxf(H) | | |
|--------|---------|-------|-------|---------|-------|-------|
| | MAC | R-MAC | R-GeM | MAC | R-MAC | R-GeM |
| Baseline | 34.6 | 40.2 | 67.3 | 14.3 | 10.5 | 44.2 |
| Cosine+$k$-nn | 48.5 | 27.6 | 53.6 | 23.6 | 76.7 | 55.3 |
| Guassian+$k$-nn | 51.7 | 27.2 | 60.1 | 28.3 | 80.0 | 61.2 |
| Diffusion+$k$-nn | **54.9** | **30.7** | 62.3 | **33.0** | 80.9 | 64.1 |
| Consine+$k$-recip | 53.3 | 30.6 | 58.8 | 27.0 | 77.7 | 58.8 |
| Guassian+$k$-recip | 51.9 | 27.0 | 60.4 | 28.4 | 80.5 | 63.0 |
| Diffusion+$k$-recip | **54.9** | 30.8 | 62.5 | 33.4 | 81.1 | 64.5 |

Table 8: Ablations of GT.

| Method | ROxf(M) | | | ROxf(H) | | |
|--------|---------|-------|-------|---------|-------|-------|
| | MAC | R-MAC | R-GeM | MAC | R-MAC | R-GeM |
| Baseline | 34.6 | 40.2 | 67.3 | 14.3 | 10.1 | 44.2 |
| Euclidean | 44.6 | 49.0 | 71.7 | 19.6 | 20.5 | 51.3 |
| Cosine | 51.3 | 57.4 | 74.4 | 26.9 | 24.9 | 55.6 |
| Jaccard | 52.0 | 57.8 | 76.1 | 28.2 | 25.3 | 57.5 |
| **GT** | **54.9** | **62.5** | **81.1** | **30.8** | **33.4** | **64.5** |

Table 9: Effect of parameter $\theta$.

| $\theta$ | 0.05 | 0.1 | 0.2 | 0.3 | 0.4 | 0.5 | 0.6 | 0.7 |
|----------|------|-----|-----|-----|-----|-----|-----|-----|
| ROxf(M) | 80.8 | 80.8 | 81.0 | 81.1 | 81.4 | 81.6 | 81.9 | **82.0** |
| ROxf(H) | 63.8 | 64.2 | **64.6** | 64.5 | 64.5 | 64.3 | 64.1 | 63.5 |

Table 10: Effect of parameter $k_2$.

| $k_2$ | 2 | 3 | 4 | 5 | 6 | 7 | 8 | 9 |
|-------|---|---|---|---|---|---|---|---|
| ROxf(M) | 77.6 | 78.9 | 78.2 | 80.2 | 80.0 | **81.1** | 80.6 | 80.2 |
| ROxf(H) | 58.2 | 59.6 | 59.6 | 60.9 | 61.3 | **64.5** | 63.2 | 62.4 |

effectiveness in capturing global-aware information. Moreover, we also encode the distributions with $k$-nn neighbors, experiments indicate that mutual information brought by $k$-reciprocal neighbors can result in a more stable performance.

**Ablations of RBR:** RBR aims to integrate the local neighbor distributions into a robust barycenter. The concept of utilizing neighborhoods to enhance robustness has been early discussed by Chum et al. (2007), which simply averages the local neighbors (NGT-AQE). Compare with the baseline that only use the instance-aware distribution for neighborhood refinery, NGT-AQE surpass baseline by 4.6%/2.2% in mAP on R-GeM. After applying our proposed RBR, the mAP further increase from 75.4%/57.6% to 81.1%/64.5% in mAP on R-GeM respectively. This validates that our proposed RBR can effectively integrate the neighbor information.

**Ablations of GT:** GT measures the pairwise distance between distributions through transporting them along the geodesic path. To verify its effectiveness, we conduct Euclidean, Cosine and Jaccard metric to compute the pairwise distance between distributions for comparison. As shown in Table 8, the superior retrieval performance is achieved by GT, surpass the second best Jarccard distance by 5.0%/7.0% on R-GeM,which validates its capability to capture geometric information and provide a more robust distance measure for neighborhood refinery.

**Analysis of hyper-parameter:** The balance weight $\theta$ in Equation 16 aims to maintain the important relationship within the Euclidean space for neighborhood refinery. The results in Table 9 validate that a $\theta$ value around 0.5 yields robust outcomes. For the parameter $k_2$ used to integrate local neighbors into a barycenter, superior performance of 81.1%/64.5% on R-GeM is achieved when $k_2 = 7$ as shown in Table 10, which indicates that its selection should both ensure the quantity and quality of neighborhoods. More analysis of other hyper-parameters $\mu$, $\sigma$, $k_1$ and $\omega$ are exhibited in Appendix E.

## 5 CONCLUSION

Searching for the most informative neighbors within the data manifold is a crucial problem in the field of machine learning and computer vision. In this paper, we propose a novel Neighbor-aware Geodesic Transportation strategy to address the problem, which consists of Manifold-aware Neighbor Encoding, Regularized Barycenter Refinery and Geodesic Transportation. Extensive evaluations on several tasks, such as re-ranking and deep clustering, demonstrate the superiority of our proposed NGT. In future work, we will leverage intermediate properties of the transport process to reduce unnecessary computations, thereby improving the efficiency of the neighborhood refinery algorithm.

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

## A BIDIRECTIONAL SIMILARITY DIFFUSION PROCESS

Given a set of feature embeddings, a sparse adjacency graph can be constructed by connecting the nearest neighbors with weighted edges, which contains the deeper manifold information. The Bidirectional Similarity Diffusion Process is designed to mine such manifold information by utilizing the adjacent weights to constrain the similarity between neighbors in a diffusion manner. Different from Bai et al. (2019); Yang et al. (2013); Zhou et al. (2012), we do not take the higher order information into consideration. Since our goal is to encode each instance into a sparse feature in the manifold space by assigning each item with a diffusion-based weight and the approximation may be influenced by outliers, using direct neighbors to regularize the weights is more robust than exploring the hypergraph. Intuitively, we introduce a reverse smooth term to maintain the symmetry of the similarity matrix. Specifically, both the forward and reverse part of similarity, $i.e.$, $\boldsymbol{F}_{ki}$ and $\boldsymbol{F}_{kj}$, $\boldsymbol{F}_{ik}$ and $\boldsymbol{F}_{jk}$, are constrained close with the same affinity weight $\boldsymbol{W}_{ij}$. In this section, we will extend the optimization problem into a matrix form and demonstrate it can be solved in an iterative way. The objective function is defined as follows:

$$\min_{\boldsymbol{F}} \underbrace{\frac{1}{4}\sum_{k=1}^{n}\sum_{i,j=1}^{n}\left(\boldsymbol{W}_{ij}\Big(\frac{\boldsymbol{F}_{ki}}{\sqrt{\boldsymbol{D}_{ii}}}-\frac{\boldsymbol{F}_{kj}}{\sqrt{\boldsymbol{D}_{jj}}}\Big)^2 + \boldsymbol{W}_{ij}\Big(\frac{\boldsymbol{F}_{ik}}{\sqrt{\boldsymbol{D}_{ii}}}-\frac{\boldsymbol{F}_{jk}}{\sqrt{\boldsymbol{D}_{jj}}}\Big)^2\right)}_{\text{smoothness}} + \underbrace{\mu\|\boldsymbol{F}-\boldsymbol{E}\|_F^2}_{\text{regularization}}, \quad (17)$$

where the left side of the expression is referred to as the smoothness term and the right side is named as the regularization term. Specifically, in the regularization term, $\boldsymbol{E}$ is introduced to restrict $\boldsymbol{F}$ from being excessively smooth, the choice of $\boldsymbol{E}$ could be a diagonal matrix $\boldsymbol{I}$ or approximated by the adjacent relationship like $(\boldsymbol{W}+\boldsymbol{W}^\top)/2$. For the convenience of the following derivation, a new identity matrix $\boldsymbol{I}$ is added, such that the smoothness term turns into:

$$\frac{1}{4}\sum_{k,l=0}^{n}\sum_{i,j=0}^{n}\left(\boldsymbol{W}_{ij}\boldsymbol{I}_{kl}\Big(\frac{\boldsymbol{F}_{ki}}{\sqrt{\boldsymbol{D}_{ii}}}-\frac{\boldsymbol{F}_{lj}}{\sqrt{\boldsymbol{D}_{jj}}}\Big)^2 + \boldsymbol{I}_{kl}\boldsymbol{W}_{ij}\Big(\frac{\boldsymbol{F}_{ik}}{\sqrt{\boldsymbol{D}_{ii}}}-\frac{\boldsymbol{F}_{jl}}{\sqrt{\boldsymbol{D}_{jj}}}\Big)^2\right). \quad (18)$$

To help refine the optimization problem into a matrix format, we introduce the vectorization operator $vec(\cdot)$ which can stack the columns in a matrix one after another to formulate a single column vector, and the kronecker product $\otimes$ which combines two matrices to produce a new one. By taking advantage of these two transformations, define the kronecker product $\mathbb{W}^{(1)} = \boldsymbol{W}\otimes\boldsymbol{I}$, $\mathbb{D}^{(1)} = \boldsymbol{D}\otimes\boldsymbol{I}$ for the former part, and $\mathbb{W}^{(2)} = \boldsymbol{I}\otimes\boldsymbol{W}$, $\mathbb{D}^{(2)} = \boldsymbol{I}\otimes\boldsymbol{D}$ for the latter part. The corresponding items between the original and matrix formation are associated with the newly defined corner markers $\alpha \equiv n(i-1)+k$, $\beta \equiv n(j-1)+l$, $\gamma \equiv n(k-1)+i$ and $\delta \equiv n(l-1)+j$. In addition to this, define the normalized matrix as $\boldsymbol{S} = \boldsymbol{D}^{-1/2}\boldsymbol{W}\boldsymbol{D}^{-1/2}$, $\mathbb{S}^{(1)} = \boldsymbol{S}\otimes\boldsymbol{I}$ and $\mathbb{S}^{(2)} = \boldsymbol{I}\otimes\boldsymbol{S}$. The following facts can be easily established:

1. $vec(\boldsymbol{F})_\alpha = \boldsymbol{F}_{ki}$ and $vec(\boldsymbol{F})_\beta = \boldsymbol{F}_{lj}$; $vec(\boldsymbol{F})_\gamma = \boldsymbol{F}_{ik}$ and $vec(\boldsymbol{F})_\delta = \boldsymbol{F}_{jl}$.

2. $\mathbb{W}^{(1)}_{\alpha\beta} = \boldsymbol{W}_{ij}\boldsymbol{I}_{kl}$, $\mathbb{D}^{(1)}_{\alpha\alpha} = \boldsymbol{D}_{ii}$ and $\mathbb{D}^{(1)}_{\beta\beta} = \boldsymbol{D}_{jj}$; $\mathbb{W}^{(2)}_{\gamma\delta} = \boldsymbol{I}_{kl}\boldsymbol{W}_{ij}$, $\mathbb{D}^{(2)}_{\gamma\gamma} = \boldsymbol{D}_{ii}$ and $\mathbb{D}^{(2)}_{\delta\delta} = \boldsymbol{D}_{jj}$.

3. $\sum_{\beta=1}^{n^2}\mathbb{W}^{(1)}_{\alpha\beta} = \mathbb{D}^{(1)}_{\alpha\alpha}$ and $\sum_{\alpha=1}^{n^2}\mathbb{W}^{(1)}_{\alpha\beta} = \mathbb{D}^{(1)}_{\beta\beta}$; $\sum_{\delta=1}^{n^2}\mathbb{W}^{(2)}_{\gamma\delta} = \mathbb{D}^{(2)}_{\gamma\gamma}$ and $\sum_{\gamma=1}^{n^2}\mathbb{W}^{(2)}_{\gamma\delta} = \mathbb{D}^{(2)}_{\delta\delta}$ since

$$\sum_{\beta=1}^{n^2}\mathbb{W}^{(1)}_{\alpha\beta} = \sum_{j=1}^{n}\boldsymbol{W}_{ij}\sum_{l=1}^{n}\boldsymbol{I}_{kl} = \boldsymbol{D}_{ii} \qquad \sum_{\alpha=1}^{n^2}\mathbb{W}^{(1)}_{\alpha\beta} = \sum_{i=1}^{n}\boldsymbol{W}_{ij}\sum_{k=1}^{n}\boldsymbol{I}_{kl} = \boldsymbol{D}_{jj}$$

$$\sum_{\delta=1}^{n^2}\mathbb{W}^{(2)}_{\gamma\delta} = \sum_{l=1}^{n}\boldsymbol{I}_{kl}\sum_{j=1}^{n}\boldsymbol{W}_{ij} = \boldsymbol{D}_{ii} \qquad \sum_{\gamma=1}^{n^2}\mathbb{W}^{(2)}_{\gamma\delta} = \sum_{k=1}^{n}\boldsymbol{I}_{kl}\sum_{i=1}^{n}\boldsymbol{W}_{ij} = \boldsymbol{D}_{jj}$$

4. $\mathbb{S}^{(1)} = (\mathbb{D}^{(1)})^{-1/2}\mathbb{W}^{(1)}(\mathbb{D}^{(1)})^{-1/2}$ and $\mathbb{S}^{(2)} = (\mathbb{D}^{(2)})^{-1/2}\mathbb{W}^{(2)}(\mathbb{D}^{(2)})^{-1/2}$ since

$$\begin{aligned} \mathbb{S}^{(1)}_{\alpha\beta} &= \boldsymbol{S}_{ij}\boldsymbol{I}_{kl} = \boldsymbol{D}_{ii}^{-1/2}\boldsymbol{W}_{ij}\boldsymbol{D}_{jj}^{-1/2}\boldsymbol{I}_{kl} \\ &= \boldsymbol{D}_{ii}^{-1/2}\boldsymbol{W}_{ij}\boldsymbol{I}_{kl}\boldsymbol{D}_{jj}^{-1/2} \\ &= (\mathbb{D}^{(1)}_{\alpha\alpha})^{-1/2}(\mathbb{W}^{(1)})_{\alpha\beta}(\mathbb{D}^{(1)}_{\beta\beta})^{-1/2} \end{aligned} \qquad \begin{aligned} \mathbb{S}^{(2)}_{\gamma\delta} &= \boldsymbol{I}_{kl}\boldsymbol{S}_{ij} = \boldsymbol{I}_{kl}\boldsymbol{D}_{ii}^{-1/2}\boldsymbol{W}_{ij}\boldsymbol{D}_{jj}^{-1/2} \\ &= \boldsymbol{D}_{ii}^{-1/2}\boldsymbol{I}_{kl}\boldsymbol{W}_{ij}\boldsymbol{D}_{jj}^{-1/2} \\ &= (\mathbb{D}^{(2)}_{\gamma\gamma})^{-1/2}\mathbb{W}^{(2)}_{\gamma\delta}(\mathbb{D}^{(2)}_{\delta\delta})^{-1/2} \end{aligned}$$

Substituting the above transformations into the smoothness term, such that Equation 18 can be reformulated into the matrix format as below:

$$
\frac{1}{4} \sum_{\alpha,\beta=1}^{n^2} \mathbb{W}_{\alpha\beta}^{(1)} \Big( \frac{vec(\boldsymbol{F})_\alpha}{\sqrt{\mathbb{D}_{\alpha\alpha}^{(1)}}} - \frac{vec(\boldsymbol{F})_\beta}{\sqrt{\mathbb{D}_{\beta\beta}^{(1)}}} \Big)^2 + \frac{1}{4} \sum_{\gamma,\delta=1}^{n^2} \mathbb{W}_{\gamma\delta}^{(2)} \Big( \frac{vec(\boldsymbol{F})_\gamma}{\sqrt{\mathbb{D}_{\gamma\gamma}^{(2)}}} - \frac{vec(\boldsymbol{F})_\delta}{\sqrt{\mathbb{D}_{\delta\delta}^{(2)}}} \Big)^2
$$

$$
= \frac{1}{4} \sum_{\alpha,\beta=1}^{n^2} \mathbb{W}_{\alpha\beta}^{(1)} \frac{vec(\boldsymbol{F})_\alpha^2}{\mathbb{D}_{\alpha\alpha}^{(1)}} + \frac{1}{4} \sum_{\alpha,\beta=1}^{n^2} \mathbb{W}_{\alpha\beta}^{(1)} \frac{vec(\boldsymbol{F})_\beta^2}{\mathbb{D}_{\beta\beta}^{(1)}} - \frac{1}{2} \sum_{\alpha,\beta=1}^{n^2} vec(\boldsymbol{F})_\alpha \frac{\mathbb{W}_{\alpha\beta}^{(1)}}{\sqrt{\mathbb{D}_{\alpha\alpha}^{(1)}\mathbb{D}_{\beta\beta}^{(1)}}} vec(\boldsymbol{F})_\beta
$$

$$
\frac{1}{4} \sum_{\gamma,\delta=1}^{n^2} \mathbb{W}_{\gamma\delta}^{(2)} \frac{vec(\boldsymbol{F})_\gamma^2}{\mathbb{D}_{\gamma\gamma}^{(2)}} + \frac{1}{4} \sum_{\gamma,\delta=1}^{n^2} \mathbb{W}_{\gamma\delta}^{(2)} \frac{vec(\boldsymbol{F})_\delta^2}{\mathbb{D}_{\delta\delta}^{(2)}} - \frac{1}{2} \sum_{\gamma,\delta=1}^{n^2} vec(\boldsymbol{F})_\gamma \frac{\mathbb{W}_{\gamma\delta}^{(2)}}{\sqrt{\mathbb{D}_{\gamma\gamma}^{(2)}\mathbb{D}_{\delta\delta}^{(2)}}} vec(\boldsymbol{F})_\delta \tag{19}
$$

$$
= vec(\boldsymbol{F})^\top \big( \mathbb{I} - \frac{1}{2}(\mathbb{D}^{(1)})^{-1/2}\mathbb{W}^{(1)}(\mathbb{D}^{(1)})^{-1/2} - \frac{1}{2}(\mathbb{D}^{(2)})^{-1/2}\mathbb{W}^{(2)}(\mathbb{D}^{(2)})^{-1/2} \big) vec(\boldsymbol{F})
$$

$$
= vec(\boldsymbol{F})^\top \big( \mathbb{I} - \frac{1}{2}\mathbb{S}^{(1)} - \frac{1}{2}\mathbb{S}^{(2)} \big) vec(\boldsymbol{F}).
$$

Basically, the regularization term is equivalent to the $l_2$-norm of $vec(\boldsymbol{F}-\boldsymbol{E})$, combine the smoothness and regularization term together, the objective function Equation 17 can be rewritten into:

$$
\min_{\boldsymbol{F}} vec(\boldsymbol{F})^T \big( \mathbb{I} - \frac{1}{2}\mathbb{S}^{(1)} - \frac{1}{2}\mathbb{S}^{(2)} \big) vec(\boldsymbol{F}) + \mu\|vec(\boldsymbol{F}-\boldsymbol{E})\|_2^2. \tag{20}
$$

**Lemma 1** *Let $\boldsymbol{A} \in \mathbb{R}^{n \times n}$, the spectral radius of $\boldsymbol{A}$ is denoted as $\rho(\boldsymbol{A}) = \max\{|\lambda|, \lambda \in \sigma(\boldsymbol{A})\}$, where $\sigma(\boldsymbol{A})$ is the spectrum of $\boldsymbol{A}$ that represents the set of all the eigenvalues. Let $\|\cdot\|$ be a matrix norm on $\mathbb{R}^{n \times n}$, given a square matrix $\boldsymbol{A} \in \mathbb{R}^{n \times n}$, $\lambda$ is an arbitrary eigenvalue of $\boldsymbol{A}$, then we have $|\lambda| \leq \rho(\boldsymbol{A}) \leq \|\boldsymbol{A}\|$.*

**Lemma 2** *Let $\boldsymbol{A} \in \mathbb{R}^{m \times m}$, $\boldsymbol{B} \in \mathbb{R}^{n \times n}$, denote $\{\lambda_i, \boldsymbol{x}_i\}_{i=1}^m$ and $\{\mu_i, \boldsymbol{y}_i\}_{i=1}^n$ as the eigen-pairs of $\boldsymbol{A}$ and $\boldsymbol{B}$ respectively. The set of $mn$ eigen-pairs of $\boldsymbol{A} \otimes \boldsymbol{B}$ is given by*

$$
\{\lambda_i\mu_j, \boldsymbol{x}_i \otimes \boldsymbol{y}_j\}_{i=1,\dots,m,\ j=1,\dots n}.
$$

Suppose the objective in Equation 20 that needs to be minimized is $J$. To prove that $J$ is convex, it is equivalent to show that its Hessian matrix $\boldsymbol{H}$ is positive. To get started, we first consider the matrix $\boldsymbol{D}^{-1}\boldsymbol{W}$, whose induced $l_\infty$-norm is equal to 1, *i.e.*, $\|\boldsymbol{D}^{-1}\boldsymbol{W}\|_\infty = 1$, since the $i$-th diagonal element in matrix $\boldsymbol{D}$ equal to the sum of the corresponding $i$-th row in matrix $\boldsymbol{W}$. Lemma 1 gives that $\rho(\boldsymbol{D}^{-1}\boldsymbol{W}) \leq 1$. As for the matrix $\boldsymbol{S} = \boldsymbol{D}^{-1/2}\boldsymbol{W}\boldsymbol{D}^{-1/2}$ we are concerned about, since we can rewrite it as $\boldsymbol{D}^{1/2}\boldsymbol{D}^{-1}\boldsymbol{W}\boldsymbol{D}^{-1/2}$, thus it is similar to $\boldsymbol{D}^{-1}\boldsymbol{W}$, *i.e.*, $\boldsymbol{S} \sim \boldsymbol{D}^{-1}\boldsymbol{W}$. Which implies that the two matrices share the same eigenvalues, such that $\rho(\boldsymbol{S}) \leq 1$. By applying Lemma 2, we can conclude that both the spectral radius of the kronecker product $\mathbb{S}^{(1)} = \boldsymbol{S} \otimes \boldsymbol{I}$ and $\mathbb{S}^{(2)} = \boldsymbol{I} \otimes \boldsymbol{S}$ is no larger than 1, *i.e.*, $\rho(\mathbb{S}^{(1)}) \leq 1$, $\rho(\mathbb{S}^{(2)}) \leq 1$.

The Hessian matrix $\boldsymbol{H}$ of Equation 20 is $2(\mu+1)\mathbb{I} - \bar{\mathbb{S}}^{(1)} - \bar{\mathbb{S}}^{(2)}$, where $2\bar{\mathbb{S}}^{(1)} = \mathbb{S}^{(1)} + (\mathbb{S}^{(1)})^\top$ and $2\bar{\mathbb{S}}^{(2)} = \mathbb{S}^{(2)} + (\mathbb{S}^{(2)})^\top$. Since we have $\mu > 0$ and $\rho(\mathbb{S}) \leq 1$, such that the eigenvalue of $\boldsymbol{H}$ is larger than 0, which means the Hessian matrix $\boldsymbol{H}$ is positive-definite and the objective function is convex. To find the optimal result of Equation 20, we can take the partial derivative of $vec(\boldsymbol{F})$:

$$
\nabla_{vec(\boldsymbol{F})} J = (2\mathbb{I} - \bar{\mathbb{S}}^{(1)} - \bar{\mathbb{S}}^{(2)})vec(\boldsymbol{F}) + 2\mu(vec(\boldsymbol{F}-\boldsymbol{E})). \tag{21}
$$

The optimal solution $\boldsymbol{F}^*$ is the root to the equation when the above partial derivative is equal to 0, we can solve the optimal solution as

$$
vec(\boldsymbol{F}^*) = \frac{2\mu}{\mu+1} \Big( 2\mathbb{I} - \frac{1}{\mu+1}\bar{\mathbb{S}}^{(1)} - \frac{1}{\mu+1}\bar{\mathbb{S}}^{(2)} \Big)^{-1} vec(\boldsymbol{E}). \tag{22}
$$

A simpler result can be obtained by substituting $\alpha$ with $\frac{1}{\mu+1}$, that is:

$$
\boldsymbol{F}^* = (1-\alpha)vec^{-1}\big((\mathbb{I} - \alpha\bar{\mathbb{S}})^{-1}vec(\boldsymbol{E})\big). \tag{23}
$$

**Lemma 3** *Let $A \in \mathbb{R}^{m \times n}$, $X \in \mathbb{R}^{n \times p}$ and $B \in \mathbb{R}^{p \times q}$ respectively, then*

$$vec(\boldsymbol{A}\boldsymbol{X}\boldsymbol{B}) = (\boldsymbol{B}^\top \otimes \boldsymbol{A})vec(\boldsymbol{X}).$$

Utilizing the relationship given by Lemma 3, we could put all the matrices in Equation 21 into the $vec(\cdot)$ operator. Additionally, set the derivative to be 0 and we can obtain that:

$$2\boldsymbol{F} - \boldsymbol{F}\bar{\boldsymbol{S}} - \bar{\boldsymbol{S}}\boldsymbol{F} + 2\mu(\boldsymbol{F} - \boldsymbol{E}) = 0. \tag{24}$$

By making some small changes to the above formula, the optimum result $\boldsymbol{F}^*$ is actually the solution to the following Lyapunov equation:

$$(\boldsymbol{I} - \alpha\bar{\boldsymbol{S}})\boldsymbol{F} + \boldsymbol{F}(\boldsymbol{I} - \alpha\bar{\boldsymbol{S}}) = 2(1 - \alpha)\boldsymbol{E}. \tag{25}$$

Directly solving this equation incurs a significantly high time complexity, but we can approximate the optimal solution at a lower cost in an iterative manner. Next, we will prove that the optimal result can be infinitely approached by the following iterative function:

$$\boldsymbol{F}^{(t+1)} = \frac{1}{2}\alpha\boldsymbol{F}^{(t)}\bar{\boldsymbol{S}}^\top + \frac{1}{2}\alpha\bar{\boldsymbol{S}}\boldsymbol{F}^{(t)} + (1 - \alpha)\boldsymbol{E}, \tag{26}$$

where $\boldsymbol{S} = \boldsymbol{D}^{-1/2}\boldsymbol{W}\boldsymbol{D}^{1/2}$, and $\bar{\boldsymbol{S}} = (\boldsymbol{S} + \boldsymbol{S}^\top)/2$. By applying Lemma 3, we can add a $vec(\cdot)$ operator on both side and rewrite the iteration process as:

$$\begin{aligned} vec(\boldsymbol{F}^{(t+1)}) &= \frac{1}{2}\alpha(\bar{\boldsymbol{S}} \otimes \boldsymbol{I})vec(\boldsymbol{F}^{(t)}) + \frac{1}{2}\alpha(\boldsymbol{I} \otimes \bar{\boldsymbol{S}})vec(F^{(t)}) + (1 - \alpha)vec(\boldsymbol{E}) \\ &= \alpha\bar{\mathbb{S}}vec(\boldsymbol{F}^{(t)}) + (1 - \alpha)vec(\boldsymbol{E}). \end{aligned} \tag{27}$$

Suppose the iteration starts from an initial value of $\boldsymbol{F}^{(0)}$, *e.g.*, $\boldsymbol{F}^{(0)}$ can be equal to the diagonal matrix $\boldsymbol{I}$ or the regularization matrix $\boldsymbol{E}$. Recursively bringing the current value into the iterative formula, we can obtain a new expression below that $\boldsymbol{F}^{(t+1)}$ is only related to the initial value $\boldsymbol{F}^{(0)}$, normalized matrix $\bar{\mathbb{S}}$ and regularization matrix $\boldsymbol{E}$, rather than dependent on the previous value $\boldsymbol{F}^{(t)}$. In formal terms, the value of matrix $\boldsymbol{F}$ after $t$-th iterations can be express as:

$$vec(\boldsymbol{F}^{(t)}) = (\alpha\bar{\mathbb{S}})^t vec(\boldsymbol{F}^{(0)}) + (1 - \alpha)\sum_{i=0}^{t-1}(\alpha\bar{\mathbb{S}})^i vec(\boldsymbol{E}). \tag{28}$$

**Lemma 4** *Let $\boldsymbol{A} \in \mathbb{R}^{n \times n}$, then $\lim_{k \to \infty} \boldsymbol{A}^k = 0$ if and only if $\rho(\boldsymbol{A}) < 1$.*

**Lemma 5** *Given a matrix $\boldsymbol{A} \in \mathbb{R}^{n \times n}$ and $\rho(\boldsymbol{A}) < 1$, the Neumann series $\boldsymbol{I} + \boldsymbol{A} + \boldsymbol{A}^2 + \cdots$ converges to $(\boldsymbol{I} - \boldsymbol{A})^{-1}$.*

Since we have already shown that the spectral radius of $\bar{\mathbb{S}}$ is no larger than 1, by taking advantage of these above two lemmas, we can easily demonstrate that the following two expressions hold true:

$$\lim_{t \to \infty}(\alpha\bar{\mathbb{S}})^t = 0, \tag{29}$$

$$\lim_{t \to \infty}\sum_{i=0}^{t-1}(\alpha\bar{\mathbb{S}})^i = (\mathbb{I} - \alpha\bar{\mathbb{S}})^{-1}. \tag{30}$$

Therefore, the iteration induce to

$$vec(F^*) = (1 - \alpha)(\mathbb{I} - \alpha\bar{\mathbb{S}})^{-1}vec(E). \tag{31}$$

By taking the inverse operator $vec^{-1}(\cdot)$ on both side, we can obtain that

$$\boldsymbol{F}^* = (1 - \alpha)vec^{-1}\big((\mathbb{I} - \alpha\bar{\mathbb{S}})^{-1}vec(\boldsymbol{E})\big). \tag{32}$$

The above expression is identical to Equation 23, which implies that the time complexity of solving the Lyapunov equation in Equation 25 can be receded to $\mathcal{O}(kn^3)$, where $k$ represents the number of iterations and $n$ denotes the dimension of matrix. Inspired by Iscen et al. (2017; 2018), the convergence rate can be further accelerated with the conjugate gradient method. In other words, the solution to the equation can be estimated with fewer iterations following Algorithm 1. Specifically, starts from an initial estimation $\boldsymbol{F}^{(0)}$, the iteration in the Bidirectional Similarity Diffusion Process will cease when the maximum count $maxiter$ is reached or the norm of the residue is less than a predefined tolerance $\delta$.

---

**Algorithm 1** Bidirectional Similarity Diffusion Process

---

**Input:** initial estimation $\boldsymbol{F}^{(0)}$, normalized kronecker matrix $\bar{\mathbb{S}}$, identity matrix $\mathbb{I}$ with the same dimension, max number of iterations $maxiter$, parameter $\alpha$, tolerance $\delta$.
  1: initialize $\boldsymbol{P}^{(0)}$ and $\boldsymbol{R}^{(0)}$ with $2(1-\alpha)\boldsymbol{E} - (\boldsymbol{I} - \alpha\bar{\boldsymbol{S}})\boldsymbol{F}^{(0)} - \boldsymbol{F}^{(0)}(\boldsymbol{I} - \alpha\bar{\boldsymbol{S}})$
  2: denote $\boldsymbol{f}_t = vec(\boldsymbol{F}^{(t)})$, $\boldsymbol{r}_t = vec(\boldsymbol{R}^{(t)})$, $\boldsymbol{p}_t = vec(\boldsymbol{P}^{(t)})$
  3: **for** $t = 0, 1, \ldots, maxiter$ **do**
  4:     compute parameter $\alpha_t = \dfrac{\boldsymbol{r}_t^\top \boldsymbol{r}_t}{2\boldsymbol{p}_t^\top(\mathbb{I} - \alpha\bar{\mathbb{S}})\boldsymbol{p}_t}$
  5:     refresh $\boldsymbol{f}_{t+1} = \boldsymbol{f}_t + \alpha_t\boldsymbol{p}_t$
  6:     update residue $\boldsymbol{r}_{t+1} = \boldsymbol{r}_t - 2\alpha_t(\mathbb{I} - \alpha\bar{\mathbb{S}})\boldsymbol{p}_t$
  7:     **if** $\|\boldsymbol{r}_{t+1}\| < \delta$ **then**
  8:         **return** $\boldsymbol{F}^* = vec^{-1}(\boldsymbol{f})$
  9:     **end if**
 10:     compute parameter $\beta_t = \dfrac{\boldsymbol{r}_{t+1}^\top \boldsymbol{r}_{t+1}}{\boldsymbol{r}_t^\top \boldsymbol{r}_t}$
 11:     refresh $\boldsymbol{p}_{t+1} = \boldsymbol{r}_{t+1} + \beta_t\boldsymbol{p}_t$
 12: **end for**
**Output:** $\boldsymbol{F}^* = vec^{-1}(\boldsymbol{f})$.

---

## B  REGULARIZED BARYCENTER REFINERY

Denote a collection containing $n$ distributions as $\mathcal{H} = \{\boldsymbol{h}_1, \boldsymbol{h}_2, \ldots, \boldsymbol{h}_n\} \in \mathbb{R}^n$, the goal of Regularized Barycenter Refinery is to find a balance centroid $\boldsymbol{a}$ of these distributions, where the weight of each distribution contributes to the calculation is given by a set $\lambda = \{\lambda_1, \lambda_2, \ldots, \lambda_n\}$. Without loss of generality, we explicitly restrict that the summation of $\lambda_s$ is equal to 1, *i.e.*, $\sum_s \lambda_s = 1$. Inspired by Peyré & Cuturi (2020); Vallender (1974); Cuturi (2013), we aim to find a center that minimizes the weighted Wasserstein distance to the distributions in the collection. In our construction, the manifold structural information is embedded in an adjacency graph, through which the distributions in the set $\mathcal{H}$ are encoded. In order to fully incorporate the manifold information when computing the Wasserstein center, we can design a cost matrix $\boldsymbol{C}$ shared by all the distributions based on the graph structure. Alternatively, it is also feasible to choose a simple 0-1 matrix as the cost matrix, *i.e.*, $\boldsymbol{C}_{ij} = \|\delta_i - \delta_j\|$, which can offer a stable performance in different situations. However, solely minimizing the weighted sum of Wasserstein distance may overlook certain dimensions that are crucial for some distributions in $\mathcal{H}$. To address this issue, we introduce an additional Euclidean distance term between the center and the distribution set $\mathcal{H}$ as a constraint, the weighted sum of Wasserstein distance and Euclidean distance are jointly minimized, weighted by $\omega$ and $1 - \omega$ respectively. Thus, the optimization function to the Regularized Barycenter Refinery problem is presented as:

$$\min_{\boldsymbol{P}, \boldsymbol{a}} \quad \sum_{s=1}^{n} \lambda_s \big(\omega\langle\boldsymbol{C}, \boldsymbol{P}_s\rangle + (1-\omega)\langle\boldsymbol{a} - \boldsymbol{h}_s, \boldsymbol{a} - \boldsymbol{h}_s\rangle\big) \tag{33}$$
$$\text{s.t.} \quad \boldsymbol{P}_s\mathbf{1} = \boldsymbol{a}, \boldsymbol{P}_s^\top\mathbf{1} = \boldsymbol{h}_s,$$

where the inner product operation $\langle\cdot\rangle$ denotes the summation of all the multiplication between corresponding components, and the operands can be either vectors or matrices. For an element $\boldsymbol{h}_s$ in the distribution set $\mathcal{H}$, there exists a transport strategy (Vallender, 1974) from $\boldsymbol{a}$ to it, denoted as $\boldsymbol{P}_s$, and all the strategies come together to make up the collection $\boldsymbol{P}$ as mentioned in Equation 33. Unless specified, all the matrix (vector) divisions, exponential, and logarithm operations proposed below are element-wise. Following Cuturi (2013); Agueh & Carlier (2011); Cuturi & Doucet (2014); Solomon et al. (2015), an entropy regularization term $H(\boldsymbol{P}_s) = \sum_{ij} \boldsymbol{P}_{ij}^s - \boldsymbol{P}_{ij}^s \log \boldsymbol{P}_{ij}^s$ weighted by $\varepsilon$ is added to the Wasserstein distance term, then the objective function turns into:

$$\min_{\boldsymbol{P}, \boldsymbol{a}} \quad \sum_{s=1}^{n} \omega\lambda_s\big(\langle\boldsymbol{C}, \boldsymbol{P}_s\rangle - \varepsilon H(\boldsymbol{P}_s)\big) + (1-\omega)\lambda_s\langle\boldsymbol{a} - \boldsymbol{h}_s, \boldsymbol{a} - \boldsymbol{h}_s\rangle \tag{34}$$
$$\text{s.t.} \quad \boldsymbol{P}_s\mathbf{1} = \boldsymbol{a}, \boldsymbol{P}_s^\top\mathbf{1} = \boldsymbol{h}_s.$$

In the objective function described above, the former and latter parts represent the entropy-regularized Wasserstein distance and the Euclidean distance constraint respectively. And the optimal result

$a$ is referred to as the regularized weighted barycenter. Define the Lagrange multipliers as $g = \{g_1, g_2, \ldots, g_n\}$ and $r = \{r_1, r_2, \ldots, r_n\}$ for the two equation constraints. Such that the Lagrangian function $\mathcal{L}(P, a, g, r)$ of the entropy regularized minimization problem in Equation 34 can be represented as:

$$\mathcal{L}(P, a, g, r) = \sum_{s=1}^{n} \lambda_s \omega \big( \langle C, P_s \rangle - \varepsilon H(P_s) + \langle g_s, a - P_s \mathbf{1} \rangle + \langle r_s, h_s - P_s^\top \mathbf{1} \rangle \big)$$
$$+ (1 - \omega) \lambda_s \langle a - h_s, a - h_s \rangle. \tag{35}$$

Characterize the optimal value of the primal problem as a function of the Lagrange multipliers, the corresponding dual Lagrangian function $\mathcal{D}(g, r)$ is defined as:

$$\mathcal{D}(g, r) = \inf_{P, a} \mathcal{L}(P, a, g, r). \tag{$\,$}$$

Given the strong convexity of the primal problem, the principle of strong duality holds, ensuring equivalence between the optimal solutions of the primal and dual problems. This enable us to solve the following maximization problem to find the optimal result to the original objective function:

$$\max_{g, r} \mathcal{D}(g, r) = \max_{g, r} \inf_{P, a} \mathcal{L}(P, a, g, r). \tag{36}$$

By substituting the Lagrangian function in Equation 35 into the dual problem, we can obtain:

$$\max_{g, r} \min_{P, a} \sum_{s=1}^{n} \omega \lambda_s \big( \langle C, P_s \rangle - \varepsilon H(P_s) + \langle g_s, a - P_s \mathbf{1} \rangle + \langle r_s, h_s - P_s^\top \mathbf{1} \rangle \big)$$
$$+ (1 - \omega) \lambda_s \langle a - h_s, a - h_s \rangle$$
$$= \max_{g, r} \sum_{s=1}^{n} \omega \lambda_s \big( \langle g_s, h_s \rangle + \min_{P} \underbrace{\langle C, P_s \rangle - \varepsilon H(P_s) - \langle g_s, P_s \mathbf{1} \rangle - \langle r_s, P_s^\top \mathbf{1} \rangle}_{\mathcal{L}_s^1(P_s)} \big) \tag{37}$$
$$+ \min_{a} \underbrace{\omega \langle \sum_{s=1}^{n} \lambda_s g_s, a \rangle + (1 - \omega) \sum_{s=1}^{n} \lambda_s \langle a - h_s, a - h_s \rangle}_{\mathcal{L}_s^2(P_s)}.$$

We can first solve the minimization objective function denoted as $\mathcal{L}_s^1(P_s)$ and $\mathcal{L}_s^2(P_s)$ within the max-min optimization problem. Replace $H(P_s)$ with its definition for the first one, we can derive:

$$\mathcal{L}_s^1(P_s) = \langle C, P_s \rangle + \varepsilon \sum_{i,j} P_{ij}^s (\log P_{ij}^s - 1) - \langle g_s, P_s \mathbf{1} \rangle - \langle r_s, P_s^\top \mathbf{1} \rangle. \tag{38}$$

This is a convex function and the first order condition gives that:

$$\frac{\partial \mathcal{L}_s^1(P_s)}{\partial P_{ij}^s} = C_{ij} + \varepsilon \log P_{ij}^s - g_i^s - r_j^s = 0. \tag{39}$$

Through rearranging the above equation, we can obtain:

$$P_s^* = \mathrm{diag}(e^{g_s/\varepsilon}) K \mathrm{diag}(e^{r_s/\varepsilon}), \tag{40}$$

where $K$ is a variant matrix of $C$ with its element defined as $K_{ij} = e^{-C_{ij}/\varepsilon}$, and operator $\mathrm{diag}(\cdot)$ can create diagonal matrices based on the given vectors. Additionally, for the simplicity of the following discussion, define the vector $u_s = e^{g_s/\varepsilon}$ and $v_s = e^{r_s/\varepsilon}$. Incorporate the above relationship into Equation 38, it can be concluded that:

$$\mathcal{L}_s^1(P_s^*) = -\varepsilon \sum_{i,j} P_{ij}^{s*}. \tag{41}$$

The second minimization problem can be directly solved with its optimal result represented as:

$$a = \sum_{s=1}^{n} \lambda_s \Big( h_s - \frac{\omega}{2(1 - \omega)} g_s \Big). \tag{42}$$

---

**Algorithm 2** Regularized Barycenter Refinery

---

**Input:** the collection of $n$ distributions $\{\boldsymbol{h}_1, \boldsymbol{h}_2, \ldots, \boldsymbol{h}_n\}$ and their weights $\{\lambda_1, \ldots, \lambda_n\}$, the cost matrix $\boldsymbol{C}$ and its variant $\boldsymbol{K} = e^{-\boldsymbol{C}/\varepsilon}$, iteration tolerance $\delta$, balance weight $\omega$, denote $\xi = \frac{\omega}{2(1-\omega)}$, and define $\oslash$ as an element-wise division for vector.
1: initialize $\{\boldsymbol{u}_1, \boldsymbol{u}_2, \ldots, \boldsymbol{u}_n\}$, $\{\boldsymbol{v}_1, \boldsymbol{v}_2, \ldots, \boldsymbol{v}_n\}$ and $\boldsymbol{a}^{(0)}$ with $\mathbf{1}$ vector.
2: **for** $t = 0, 1, \ldots, maxiter$ **do**
3:     **for** $s = 1, 2, \ldots, n$ **do**
4:         $\boldsymbol{v}_s^{(t+1)} = \boldsymbol{h}_s \oslash \boldsymbol{K}^\top \boldsymbol{u}_s^{(t)}$
5:     **end for**
6:     $\boldsymbol{a}^{(t+1)} = \sum_s \lambda_s (\boldsymbol{h}_s - \varepsilon \xi \log \boldsymbol{u}_s^{(t)})$
7:     **if** $\|\boldsymbol{a}^{(t+1)} - \boldsymbol{a}^{(t)}\| \leq \delta$ **then**
8:         **return** $\boldsymbol{a}$
9:     **end if**
10:    **for** $s = 1, 2, \ldots, n$ **do**
11:       $\boldsymbol{u}_s^{(t+1)} = \boldsymbol{a}^{(t+1)} \oslash \boldsymbol{K} \boldsymbol{v}_s^{(t+1)}$
12:    **end for**
13: **end for**
**Output:** the regularized weighted barycenter $\boldsymbol{a}$.

---

Substitute $\boldsymbol{P}^*$ and $\boldsymbol{a}$ back into Equation 37, it becomes:

$$\max_{\boldsymbol{g}, \boldsymbol{r}} \sum_{s=1}^n \omega \lambda_s \left( \langle \boldsymbol{r}_s, \boldsymbol{h}_s \rangle - \varepsilon \sum_{i,j} e^{\boldsymbol{g}_i^s/\varepsilon} \boldsymbol{K}_{ij} e^{\boldsymbol{r}_j^s/\varepsilon} \right) - \frac{\omega^2}{4(1-\omega)} \langle \sum_{s=1}^n \lambda_s \boldsymbol{g}_s, \sum_{s=1}^n \lambda_s \boldsymbol{g}_s \rangle$$

$$+ \omega \langle \sum_{s=1}^n \lambda_s \boldsymbol{g}_s, \sum_{s=1}^n \lambda_s \boldsymbol{h}_s \rangle - (1-\omega) \langle \sum_{s=1}^n \lambda_s \boldsymbol{h}_s, \sum_{s=1}^n \lambda_s \boldsymbol{h}_s \rangle + (1-\omega) \sum_{s=1}^n \lambda_s \langle \boldsymbol{h}_s, \boldsymbol{h}_s \rangle. \tag{43}$$

With fixed $\boldsymbol{g}$, the first order condition with respect to $\boldsymbol{r}_j^s$ gives that:

$$\boldsymbol{h}_j^s - \sum_i e^{\boldsymbol{g}_i^s/\varepsilon} \boldsymbol{K}_{ij} e^{\boldsymbol{r}_j^s/\varepsilon} = 0, \tag{44}$$

by simplifying the representation with $\boldsymbol{u}_s$ and $\boldsymbol{v}_s$ and reorganizing the equation provided above, we can obtain the following relationship:

$$\text{diag}(\boldsymbol{v}_s) \boldsymbol{K}^\top \boldsymbol{u}_s = \boldsymbol{h}_s. \tag{45}$$

With fixed $\boldsymbol{r}$, the first order condition with respect to $\boldsymbol{g}_i^s$ gives that:

$$\sum_j e^{\boldsymbol{g}_i^s/\varepsilon} \boldsymbol{K}_{ij} e^{\boldsymbol{r}_j^s/\varepsilon} + \frac{\omega}{2(1-\omega)} \sum_s \lambda_s \boldsymbol{g}_i^s = \sum_s \lambda_s \boldsymbol{h}_i^s, \tag{46}$$

denote $\xi = \frac{\omega}{2(1-\omega)}$ for the sake of simplification, replace the exponential components with $\boldsymbol{u}_s$ and $\boldsymbol{v}_s$ and we can obtain that:

$$\text{diag}(\boldsymbol{u}_s) \boldsymbol{K} \boldsymbol{v}_s = \sum_{s=1}^n \lambda_s (\boldsymbol{h}_s - \varepsilon \xi \log \boldsymbol{u}_s). \tag{47}$$

Following Algorithm 2, the numerical solutions can be obtained in an iterative manner.

## C   APPROXIMATE REGULARIZED BARYCENTER REFINERY

The Algorithm 2 may suffers from the drawback of numerical instability in certain situations. To address this issue, we propose an approximate solution to find the regularized weighted barycenter. Firstly, the optimization objective $\boldsymbol{a}$ in Equation 33 is relaxed to a weighted sum of $\boldsymbol{a}_1$ and $\boldsymbol{a}_2$, *i.e.*, $\boldsymbol{a} = \omega \boldsymbol{a}_1 + (1-\omega) \boldsymbol{a}_2$. The optimal transport strategies from $\boldsymbol{a}_1$ to the distribution set $\mathcal{H}$ is denoted as $\boldsymbol{P}$, and the objective function to the Approximate Regularized Weighted problem is defined as:

$$\min_{\boldsymbol{P}, \boldsymbol{a}_1, \boldsymbol{a}_2} \quad \sum_{s=1}^n \omega \lambda_s \big( \langle \boldsymbol{C}, \boldsymbol{P}_s \rangle - \varepsilon H(\boldsymbol{P}_s) \big) + (1-\omega) \lambda_s \langle \boldsymbol{a}_2 - \boldsymbol{h}_s, \boldsymbol{a}_2 - \boldsymbol{h}_s \rangle \tag{48}$$

$$\text{s.t.} \quad \boldsymbol{P}_s \mathbf{1} = \boldsymbol{a}_1, \boldsymbol{P}_s^\top \mathbf{1} = \boldsymbol{h}_s.$$

Since $\boldsymbol{a}_1$ and $\boldsymbol{a}_2$ are independent to each other, such that we can separately optimize the values of $\boldsymbol{a}_1$ and $\boldsymbol{a}_2$. When solving the optimization function involving $\boldsymbol{a}_2$, we can obtain that $\boldsymbol{a}_2 = \sum_s \lambda_s \boldsymbol{h}_s$. And when solving the optimization problem involving the weighted sum of regularized Wasserstein distance with respect to $\boldsymbol{a}_1$, the optimization objective can be expressed as:

$$\min_{\boldsymbol{P}, \boldsymbol{a}_1} \quad \sum_{s=1}^{n} \lambda_s \big( \langle \boldsymbol{C}, \boldsymbol{P}_s \rangle - \varepsilon H(\boldsymbol{P}_s) \big) \tag{49}$$

$$\text{s.t.} \quad \boldsymbol{P}_s \boldsymbol{1} = \boldsymbol{a}_1, \boldsymbol{P}_s^\top \boldsymbol{1} = \boldsymbol{h}_s.$$

Similar to Section B, the Lagrangian function for the entropy regularized optimization problem in Equation 49 is defined as:

$$\mathcal{L}(\boldsymbol{P}, \boldsymbol{a}_1, \boldsymbol{g}, \boldsymbol{r}) = \sum_{s=1}^{n} \lambda_s \big( \langle \boldsymbol{C}, \boldsymbol{P}_s \rangle - \varepsilon H(\boldsymbol{P}_s) + \langle \boldsymbol{g}_s, \boldsymbol{a}_1 - \boldsymbol{P}_s \boldsymbol{1} \rangle + \langle \boldsymbol{r}_s, \boldsymbol{h}_s - \boldsymbol{P}_s^\top \boldsymbol{1} \rangle \big), \tag{50}$$

where $\boldsymbol{g}$ and $\boldsymbol{r}$ are the Lagrangian multipliers for the two equation constraints respectively. Since the objective function is strictly convex and such that the strong duality holds. Solving the primal problem is equivalent to finding the maximum value of the following dual Lagrangian function:

$$\mathcal{D}(\boldsymbol{g}, \boldsymbol{r}) = \inf_{\boldsymbol{P}, \boldsymbol{a}_1} \mathcal{L}(\boldsymbol{P}, \boldsymbol{a}_1, \boldsymbol{g}, \boldsymbol{r}). \tag{51}$$

Substitute the definition of $\mathcal{L}(\boldsymbol{P}, \boldsymbol{a}_1, \boldsymbol{g}, \boldsymbol{r})$ into the dual Lagrangian function $\mathcal{D}(\boldsymbol{g}, \boldsymbol{r})$, and the maximize objective function can be reorganized as:

$$\max_{\boldsymbol{g}, \boldsymbol{r}} \min_{\boldsymbol{P}, \boldsymbol{a}_1} \sum_{s=1}^{n} \lambda_s \big( \langle \boldsymbol{C}, \boldsymbol{P}_s \rangle + \varepsilon H(\boldsymbol{P}_s) + \langle \boldsymbol{g}_s, \boldsymbol{a}_1 - \boldsymbol{P}_s \boldsymbol{1} \rangle + \langle \boldsymbol{r}_s, \boldsymbol{h}_s - \boldsymbol{P}_s^\top \boldsymbol{1} \rangle \big)$$

$$= \max_{\boldsymbol{g}, \boldsymbol{r}} \sum_{s=1}^{n} \lambda_s \big( \langle \boldsymbol{g}_s, \boldsymbol{b}_s \rangle + \min_{\boldsymbol{P}} \langle \boldsymbol{C}, \boldsymbol{P}_s \rangle + \varepsilon H(\boldsymbol{P}_s) - \langle \boldsymbol{g}_s, \boldsymbol{P}_s \boldsymbol{1} \rangle - \langle \boldsymbol{r}_s, \boldsymbol{P}_s^\top \boldsymbol{1} \rangle \big) \tag{52}$$

$$+ \min_{\boldsymbol{a}_1} \big\langle \sum_{s=1}^{n} \lambda_s \boldsymbol{g}_s, \boldsymbol{a}_1 \big\rangle.$$

The second minimization objective leads to $\sum_s \lambda_s \boldsymbol{g}_s = 0$, otherwise, there exists an $\boldsymbol{a}_1$ such that the optimal value becomes $-\infty$. Similar to Equation 38, we can define:

$$\mathcal{L}_s(\boldsymbol{P}_s) = \langle \boldsymbol{C}, \boldsymbol{P}_s \rangle + \varepsilon \sum_{i,j} \boldsymbol{P}_{ij}^s (\log \boldsymbol{P}_{ij}^s - 1) - \langle \boldsymbol{g}_s, \boldsymbol{P}_s \boldsymbol{1} \rangle - \langle \boldsymbol{r}_s, \boldsymbol{P}_s^\top \boldsymbol{1} \rangle. \tag{53}$$

The first order condition of $\mathcal{L}_s(\boldsymbol{P}_s)$ gives that:

$$\boldsymbol{P}_s^* = \text{diag}(e^{\boldsymbol{g}_s/\varepsilon}) \boldsymbol{K} \text{diag}(e^{\boldsymbol{r}_s/\varepsilon}). \tag{54}$$

where $\boldsymbol{K}$, defined as $\boldsymbol{K} = e^{-\boldsymbol{C}/\varepsilon}$, is a variant matrix of $\boldsymbol{C}$. The operator $\text{diag}(\cdot)$ can transform a vector into a corresponding diagonal matrix. Similar to Section B, define $\boldsymbol{u}_s = e^{\boldsymbol{g}_s/\varepsilon}$ and $\boldsymbol{v}_s = e^{\boldsymbol{r}_s/\varepsilon}$ for the simplicity of discussion, substitute $\boldsymbol{P}_s^*$ into $\mathcal{L}_s(\boldsymbol{P}_s)$, we can obtain:

$$\mathcal{L}_s(\boldsymbol{P}_s^*) = -\varepsilon \sum_{i,j} \boldsymbol{P}_{ij}^{s*}. \tag{55}$$

The optimization question is then transformed into:

$$\max_{\boldsymbol{g}, \boldsymbol{r}} \quad \sum_{s=1}^{n} \lambda_s \Big( \langle \boldsymbol{r}_s, \boldsymbol{h}_s \rangle - \varepsilon \sum_{i,j} e^{\boldsymbol{g}_i^s/\varepsilon} \boldsymbol{K}_{ij} e^{\boldsymbol{r}_j^s/\varepsilon} \Big)$$

$$\text{s.t.} \quad \sum_{s=1}^{n} \lambda_s \boldsymbol{g}_s = 0. \tag{56}$$

Define the Lagrangian function of problem 56 as:

$$\mathcal{L}(\boldsymbol{g}, \boldsymbol{r}, \boldsymbol{b}) = \sum_{s=1}^{n} \lambda_s \Big( \varepsilon \sum_{i,j} e^{\boldsymbol{g}_i^s/\varepsilon} \boldsymbol{K}_{ij} e^{\boldsymbol{r}_j^s/\varepsilon} - \langle \boldsymbol{r}_s, \boldsymbol{h}_s \rangle \Big) - \big\langle \boldsymbol{b}, \sum_{s=1}^{n} \lambda_s \boldsymbol{g}_s \big\rangle. \tag{57}$$

---

**Algorithm 3** Approximate Regularized Barycenter Refinery

---

**Input:** the collection of $n$ distributions $\{\boldsymbol{h}_1, \boldsymbol{h}_2, \ldots, \boldsymbol{h}_n\}$ and their weights $\{\lambda_1, \ldots, \lambda_n\}$, the cost matrix $\boldsymbol{C}$ and its variant $\boldsymbol{K} = e^{-\boldsymbol{C}/\varepsilon}$, iteration tolerance $\delta$, balance weight $\omega$, and define $\oslash$ as an element-wise division for vector.
1: initialize $\{\boldsymbol{u}_1, \boldsymbol{u}_2, \ldots, \boldsymbol{u}_n\}$, $\{\boldsymbol{v}_1, \boldsymbol{v}_2, \ldots, \boldsymbol{v}_n\}$ and $\boldsymbol{a}_1^{(0)}$ with $\mathbf{1}$ vector.
2: compute $\boldsymbol{a}_2 = \sum_s \lambda_s \boldsymbol{h}_s$
3: **for** $t = 0, 1, \ldots, maxiter$ **do**
4:     **for** $s = 1, 2, \ldots, n$ **do**
5:         $\boldsymbol{v}_s^{(t+1)} = \boldsymbol{h}_s \oslash \boldsymbol{K}^\top \boldsymbol{u}_s^{(t)}$
6:     **end for**
7:     $\boldsymbol{a}_1^{(t+1)} = \prod_s (\boldsymbol{K}_s \boldsymbol{v}_s^{(t+1)})^{\lambda_s}$
8:     **if** $\|\boldsymbol{a}_1^{(t+1)} - \boldsymbol{a}_1^{(t)}\| \leq \delta$ **then**
9:         **return** $\boldsymbol{a} = \omega \boldsymbol{a}_1 + (1 - \omega) \boldsymbol{a}_2$
10:     **end if**
11:     **for** $s = 1, 2, \ldots, n$ **do**
12:         $\boldsymbol{u}_s^{(t+1)} = \boldsymbol{a}_1^{(t+1)} \oslash \boldsymbol{K} \boldsymbol{v}_s^{(t+1)}$
13:     **end for**
14: **end for**
**Output:** the approximate regularized weighted barycenter $\boldsymbol{a} = \omega \boldsymbol{a}_1 + (1 - \omega) \boldsymbol{a}_2$.

---

With fixed $\boldsymbol{g}$, the first order condition with respect to $\boldsymbol{r}_j^s$ gives that:

$$\frac{\partial \mathcal{L}(\boldsymbol{g}, \boldsymbol{r}, \boldsymbol{b})}{\partial \boldsymbol{r}_j^s} = \lambda_s \sum_i e^{\boldsymbol{g}_i^s/\epsilon} \boldsymbol{K}_{ij} e^{\boldsymbol{r}_j^s/\epsilon} - \lambda_s \boldsymbol{h}_j^s = 0. \tag{58}$$

The above equation can be simplified with $\boldsymbol{u}_s$ and $\boldsymbol{v}_s$, such that the following relationship holds:

$$\operatorname{diag}(\boldsymbol{v}_s) \boldsymbol{K}^\top \boldsymbol{u}_s = \boldsymbol{h}_s. \tag{59}$$

With fixed $\boldsymbol{r}_s$, the first order condition with respect to $\boldsymbol{g}_i^s$ gives that:

$$\frac{\partial \mathcal{L}(\boldsymbol{g}, \boldsymbol{r}, \boldsymbol{b})}{\partial \boldsymbol{g}_i^s} = \lambda_s \sum_j e^{\boldsymbol{g}_i^s/\varepsilon} \boldsymbol{K}_{ij} e^{\boldsymbol{r}_j^s/\varepsilon} - \lambda_s \boldsymbol{b}_i = 0. \tag{60}$$

By utilizing the constraint that $\boldsymbol{P}_s \mathbf{1} = \boldsymbol{a}_1$, we can derive:

$$\operatorname{diag}(\boldsymbol{u}_s) \boldsymbol{K} \boldsymbol{v}_s = \boldsymbol{b} = \boldsymbol{a}_1. \tag{61}$$

Additionally, notice that the constraint $\sum_s \lambda_s \boldsymbol{g}_s = 0$ holds. By replacing $\boldsymbol{g}_s$ with $\varepsilon \log \boldsymbol{u}_s$, we can obtain that:

$$\sum_{s=1}^n \varepsilon \lambda_s \log \frac{\boldsymbol{a}_1}{\boldsymbol{K} \boldsymbol{v}_s} = 0, \tag{62}$$

such that the following relationship can be concluded:

$$\boldsymbol{a}_1 = \prod_s (\boldsymbol{K} \boldsymbol{v}_s)^{\lambda_s}. \tag{63}$$

The numerical solution of $\boldsymbol{a}_1$ can be derived in an iterative manner following Algorithm 3, while the optimal value of $\boldsymbol{a}_2$ can be directly solved. Finally, the approximate regularized weighted barycenter can be obtained by $\boldsymbol{a} = \omega \boldsymbol{a}_1 + (1 - \omega) \boldsymbol{a}_2$.

## D  NEIGHBOR-AWARE GEODESIC TRANSPORTATION FOR DEEP CLUSTERING

The proposed neighborhood refinery method NGT can also be applied to the deep clustering task in an online manner. Given a batch of images $\mathcal{B}$ sampled from the whole dataset, each of them will pass through two sets of random augmentations, denoted as $t \sim \mathcal{T}$ and $t' \sim \mathcal{T}'$. Following the training procedure defined in BYOL (Grill et al., 2020), the two sets of the augmentation images are

encoded with the online network $\mathcal{F}_o(\cdot)$ and target network $\mathcal{F}_t(\cdot)$ respectively. Moreover, an additional non-linear predictor is cascaded behind the online network, then this branch is formally defined as $q(\mathcal{F}_o(\cdot))$.

The self-supervised learning framework proposed in BYOL is a non-contrastive learning approach, which aims to enforce the similarity between a referenced feature and its corresponding feature from the other augmentation views. Such that it can be viewed as instance-aware concordance discrimination approach, which can be formulated as:

$$\mathcal{L}_I = - \mathop{\mathbb{E}}_{\boldsymbol{x}_i \in \mathcal{B}} [\langle q(\mathcal{F}_o(t(\boldsymbol{x}_i))), \mathcal{F}_t(t'(\boldsymbol{x}_i)) \rangle], \tag{64}$$

where $\langle \cdot \rangle$ is the inner product operator to calculate cosine similarity, the over-clustered representation is bootstrapped via the stop-gradient and momentum updating mechanism, which outperforms the contrastive based self-supervised method significantly.

Although BYOL has demonstrated its superiority as a pre-training task for enhancing network representations for downstream tasks, there is still room to improve its discriminative power by incorporating the diverse positive samples from local neighborhood. Such that the instance-aware concordance framework can be adapted to the group level. Based on this idea, previous works (Dwibedi et al., 2021; Koohpayegani et al., 2021; Van Gansbeke et al., 2020; Niu et al., 2022) further enhance the consistency between instance and neighbors to improve the capability of the self-supervised framework. The group-aware concordance can be formally defined as:

$$\mathcal{L}_G = - \mathop{\mathbb{E}}_{x_i \in \mathcal{B}} [ \mathop{\mathbb{E}}_{x_j \in \mathcal{N}(\boldsymbol{x}_i)} [\langle q(\mathcal{F}_o(t(\boldsymbol{x}_i))), \mathcal{F}_t(t'(\boldsymbol{x}_j)) \rangle]], \tag{65}$$

where the $\mathcal{N}(\boldsymbol{x}_i)$ denotes the set of neighborhood samples of $\boldsymbol{x}_i$ within the Euclidean space. Since $k$-nearest neighbors may unavoidably include negative samples in the computation of group-aware loss, this error could accumulate throughout the training process. Thus exploiting the robust neighbors (Yu et al., 2023; Dwibedi et al., 2021) from the data manifold constructed by the batch of image features can help improve the quality of neighborhoods and provide a more precise supervision signals. Intuitively, the original $k$-nearest neighbors can be replaced with our proposed NGT neighborhood, such that the the negative influence can be mitigated with the precise neighboring information brought by NGT. Similar to previous neighbor guided self-supervised learning strategy, our algorithm consists of two parts, the first $t_0$ epochs for instance-aware pre-training and the rest epochs for group-aware concordance training with NDT to find refined neighborhood, the overall loss is formulated as:

$$\mathcal{L}_{total} = \mathbb{1}(t < t_0)\mathcal{L}_I + \mathbb{1}(t \geq t_0)\mathcal{L}_G. \tag{66}$$

For the deep clustering tasks, the performance are measured on five widely used benchmarks, including CIFAR-10 (Krizhevsky et al., 2009), CIFAR-20 (Krizhevsky et al., 2009), STL-10 (Coates et al., 2011), ImageNet-10 (Chang et al., 2017), and ImageNet-Dogs (Chang et al., 2017). Where CIFAR-10 and CIFAR-20 are moderate scale dataset both containing 60,000 images. Following the settings in Huang et al. (2023); Yu et al. (2023), we resize the images to the scale of $32 \times 32$ for CIFAR-10 and CIFAR-20, $96 \times 96$ for STL-10 and ImageNet-10, $224 \times 224$ for ImageNet-Dogs.

We strictly follow Huang et al. (2023); Shen et al. (2021b); Yu et al. (2023); Li et al. (2021; 2022) to perform fair comparison. Specifically, we conduct the same augmentation strategies and adopt the stochastic gradient optimizer and the cosine decay learning schedule with first 50 epochs for warming up. The entire network is trained for 1000 epochs, with the initial 800 epochs utilizing standard BYOL loss $\mathcal{L}_I$, followed by the remaining 200 epochs employing $\mathcal{L}_G$, which incorporates our proposed method. In order to enhance training efficiency and mitigate the potential error accumulations, we employ a straightforward 0-1 matrix to calculate the Geodesic Transportation distance. For the architecture, we use ResNet-18 on CIFAR-10 and CIFAR-20 and ResNet-34 on the rest of datasets for a fair comparison, The base learing rate is 0.05 and batchsize we use is 128 for CIFAR-10 and 64 for others, which is smaller than the existing methods. The select the hyper parameter of $k_1 = 10$ and $k_2 = 4$ for the first four datasets, and $k_1 = 25$ and $k_2 = 8$ for the challenging ImageNet-Dogs with many categories that are not easy to distinguish. We train the model with 4 GTX 3090 GPU, the performance converges within 200 epochs of training, which takes about 4 hours. During the evaluation procedure, the labels of features encoded by the target encoder are assigned by $k$-means to validate the clustering performance for a fair comparison.

# E  EXTENDED EXPERIMENT

In this section, we provide more experiments on image retrieval tasks based on MAC and R-MAC proposed by Tolias et al. (2016), as well as some extra analysis of hyper-parameters based on R-GeM (Radenović et al., 2019) and CVNet (Lee et al., 2022).

Table 11: Evaluation of the retrieval performance based on MAC, best in **bold**.

| Method | Easy | | Medium | | Hard | |
|---|---|---|---|---|---|---|
| | *R*Oxf | *R*Par | *R*Oxf | *R*Par | *R*Oxf | *R*Par |
| MAC | 47.2 | 69.7 | 34.6 | 55.7 | 14.3 | 32.6 |
| AQE | 54.4 | 80.9 | 40.6 | 67.0 | 17.1 | 45.2 |
| $\alpha$QE | 50.3 | 77.8 | 37.1 | 64.4 | 16.3 | 43.0 |
| kNN | 56.6 | 79.7 | 41.6 | 66.5 | 17.4 | 44.5 |
| DFS | 54.6 | 83.8 | 40.6 | 74.0 | 18.8 | 58.1 |
| AQEwD | 52.8 | 79.6 | 39.7 | 65.0 | 17.3 | 42.9 |
| SG | 46.1 | 75.9 | 36.1 | 60.4 | 16.6 | 38.8 |
| RDP | 59.0 | 85.2 | 45.3 | 76.3 | 21.4 | 58.9 |
| GSS | 60.0 | 87.5 | 45.4 | 76.7 | 22.8 | 59.7 |
| STML | 61.4 | 86.8 | 46.7 | 76.9 | 22.3 | 59.5 |
| ConAff | 65.5 | 88.7 | 50.1 | 79.3 | 25.6 | 62.4 |
| **NGT** | **71.4** | **90.4** | **54.9** | **81.1** | **30.8** | **65.8** |

Table 12: Evaluation of the retrieval performance based on R-MAC, best in **bold**.

| Method | Easy | | Medium | | Hard | |
|---|---|---|---|---|---|---|
| | *R*Oxf | *R*Par | *R*Oxf | *R*Par | *R*Oxf | *R*Par |
| R-MAC | 61.2 | 79.3 | 40.2 | 63.8 | 10.1 | 38.2 |
| AQE | 69.4 | 85.7 | 47.8 | 71.1 | 15.9 | 47.9 |
| $\alpha$QE | 64.9 | 84.7 | 42.8 | 70.8 | 11.4 | 47.8 |
| kNN | 70.6 | 84.6 | 48.9 | 70.2 | 16.0 | 46.1 |
| DFS | 70.0 | 87.5 | 51.8 | 78.8 | 20.3 | 63.5 |
| AQEwD | 70.5 | 85.9 | 48.7 | 70.7 | 15.3 | 46.9 |
| SG | 60.1 | 84.9 | 42.7 | 68.4 | 16.5 | 45.4 |
| RDP | 73.7 | 88.8 | 54.3 | 79.6 | 22.2 | 61.3 |
| GSS | 75.0 | 89.9 | 54.7 | 78.5 | 24.4 | 60.5 |
| STML | 71.8 | 88.7 | 53.2 | 78.2 | 23.4 | 58.8 |
| ConAff | 77.6 | 88.0 | 56.4 | 80.0 | 27.5 | 61.3 |
| **NGT** | **84.3** | **90.0** | **62.5** | **82.9** | **33.4** | **68.4** |

Table 13: Effect of $\omega$, based on R-GeM.

| $\omega$ | 0 | 0.05 | 0.1 | 0.2 | 0.3 | 0.4 | 0.5 | 0.6 |
|---|---|---|---|---|---|---|---|---|
| *R*Oxf(M) | 81.3 | **81.4** | 81.1 | 81.2 | 81.1 | 81.0 | 80.9 | 80.7 |
| *R*Oxf(H) | 63.0 | 63.6 | 64.5 | **64.6** | 64.5 | 64.3 | 64.0 | 63.5 |

Table 16: Effect of $\omega$, based on CVNet.

| $\omega$ | 0 | 0.05 | 0.1 | 0.2 | 0.3 | 0.4 | 0.5 | 0.6 |
|---|---|---|---|---|---|---|---|---|
| *R*Oxf(M) | 88.9 | 89.0 | **89.3** | 89.2 | 89.1 | 89.0 | 88.9 | 88.8 |
| *R*Oxf(H) | 72.9 | 73.1 | **73.8** | 73.7 | 73.4 | 72.9 | 72.9 | 72.7 |

Table 14: Effect of $\mu$, based on R-GeM.

| $\mu$ | 0.1 | 0.2 | 0.3 | 0.4 | 0.5 | 0.6 | 0.7 | 0.8 |
|---|---|---|---|---|---|---|---|---|
| *R*Oxf(M) | 81.0 | **81.1** | 81.1 | 80.8 | 80.8 | 80.7 | 80.7 | 80.7 |
| *R*Oxf(H) | 64.2 | **64.5** | 64.3 | 63.6 | 63.6 | 63.4 | 63.3 | 63.2 |

Table 17: Effect of $\mu$, based on CVNet.

| $\mu$ | 0.1 | 0.2 | 0.3 | 0.4 | 0.5 | 0.6 | 0.7 | 0.8 |
|---|---|---|---|---|---|---|---|---|
| *R*Oxf(M) | 89.0 | 89.3 | 89.4 | 89.5 | **89.6** | 89.6 | 89.6 | 89.6 |
| *R*Oxf(H) | 73.4 | 73.8 | 73.9 | 74.0 | **74.2** | 74.1 | 74.2 | 74.2 |

Table 15: Effect of $\sigma$, based on R-GeM.

| $\sigma$ | 0.1 | 0.2 | 0.3 | 0.4 | 0.5 | 0.6 | 0.7 | 0.8 |
|---|---|---|---|---|---|---|---|---|
| *R*Oxf(M) | 80.5 | **81.2** | 81.1 | 80.6 | 80.2 | 79.7 | 79.4 | 79.1 |
| *R*Oxf(H) | 63.8 | **63.8** | 64.5 | 63.8 | 62.5 | 61.5 | 60.8 | 60.4 |

Table 18: Effect of $\sigma$, based on CVNet.

| $\sigma$ | 0.1 | 0.2 | 0.3 | 0.4 | 0.5 | 0.6 | 0.7 | 0.8 |
|---|---|---|---|---|---|---|---|---|
| *R*Oxf(M) | 89.0 | 88.9 | **89.3** | 89.1 | 88.9 | 88.7 | 88.7 | 88.6 |
| *R*Oxf(H) | 73.4 | 73.3 | **73.8** | 73.5 | 72.9 | 72.5 | 72.3 | 72.1 |

Table 19: Summary of the datasets.

| Dataset | Split | # Samples | # Classes | Image Size |
|---|---|---|---|---|
| CIFAR-10 | Train+Test | 60,000 | 10 | $32 \times 32$ |
| CIFAR-20 | Train+Test | 60,000 | 20 | $32 \times 32$ |
| STL-10 | Train+Test | 13,000 | 10 | $96 \times 96$ |
| ImageNet-10 | Train | 13,000 | 10 | $96 \times 96$ |
| ImageNet-Dogs | Train | 19,500 | 15 | $96 \times 96$ |

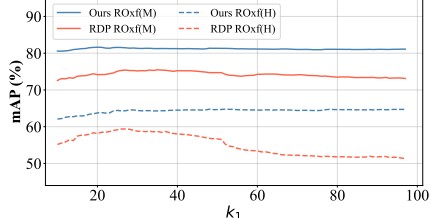

Figure 4: Effect of $k_1$, based on the image features extracted by R-GeM (Radenović et al., 2019). We plot the effect of similar hyper-parameters used for diffusion in the same figure, and the comparative results indicate that our method demonstrates higher stability.

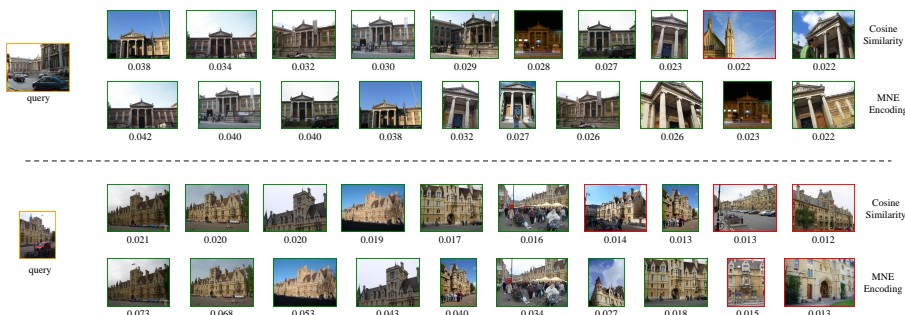

Figure 5: Quantitative analysis of the Manifold-aware Neighborhood Encoding strategy. In each query, the top row denotes the cosine similarity weight for encoding the original feature into a nonlinear space, and the bottom row represents the weight obtained by the bidirectional similarity diffusion strategy. The weights are sorted in reverse order, and we can observe that MNE can filter out important samples and assign them with higher weights.

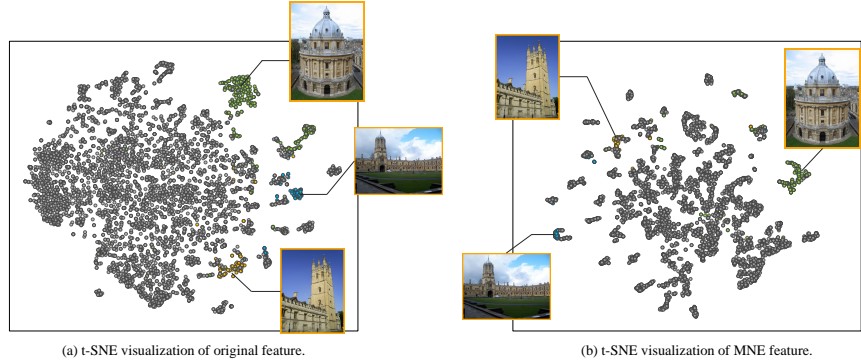

(a) t-SNE visualization of original feature.    (b) t-SNE visualization of MNE feature.

Figure 6: The t-SNE visualization of the original feature space and the MNE feature space. Compared with the original image feature, MNE can achieve better clustering performance and reduce the effect of outliers, resulting in a higher retrieval result.

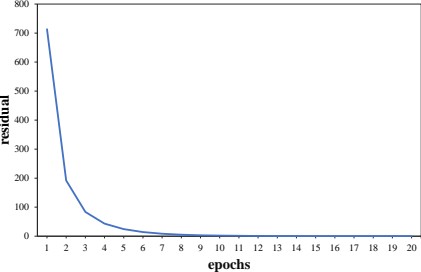

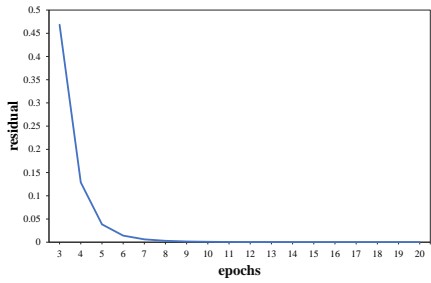

Figure 7: Convergence analysis of MNE. The residual represents the Frobenius norm of the target matrix for two iterations.

Figure 8: Convergence analysis of RBR. The residual represents the F-norm of the target matrix for two iterations.

