# OpenReview forum: "Neighbor-aware Geodesic Transportation for Neighborhood Refinery"
_ICLR.cc/2025/Conference — ICLR 2025 Conference Withdrawn Submission_

### Official Review · Reviewer_6tqL · 2024-10-30

**Soundness:** 2
**Presentation:** 1
**Contribution:** 2
**Rating:** 5
**Confidence:** 3

**Summary:**

This paper investigates the topic of neighborhood refinery. To capture global relationships and mitigate the negative impacts of outliers in the neighborhood, a method named NGT is proposed, which consists of three main components, namely Manifold-aware Neighbor Encoding
(MNE), Regularized Barycenter Refinery (RBR), and Geodesic Transportation (GT). The effectiveness of the proposed method is demonstrated on re-ranking and deep clustering tasks. However, I have concerns about its novelty and its presentation.

**Strengths:**

1. Neighborhood refinery is an important topic that benefits many downstream tasks.
2. The proposed method seems to perform well on most tasks, revealing its effectiveness. Yet, the complexity of the proposed method may limit its scalability and practical applications.

**Weaknesses:**

1. The novelty of the proposed method is limited. In particular, many concepts are ponderously used, lacking clear explanations. For example, why the similarity matrix F is manifold-aware (line 212)? Rigorous analyses are required.
2. The presentation is poor. Many notations are ambiguous or lack specifications. For example, what is $\delta$ in line 258? Is the cost matrix C in line 257 the same as the geodesic distance matrix C in line 318.
3. The details of datasets are not provided. Summarize them in a table would be helpful.
4. The effect of parameter $k_1$ is not discussed.
5. In Table 2, the best result on Hard ROxf is not achieved by your method, which has been mistakenly highlighted in bold.
6. The method is claimed to reduce the negative impact of outliers in neighborhoods, but the authors do not provide relevant experimental results to support their claims.

**Questions:**

See weaknesses.

---

> ### Author Response · Authors · 2024-11-24
> **Thank you for your valuable feedbacks.**
>
> 1. **More Explanation to the MNE Module**
>
> Manifold learning assumes that high-dimensional data lies on a lower-dimensional manifold. **The goal is to learn this manifold's structure to perform tasks like retrieval while respecting the data's intrinsic geometry**. In machine learning, particularly in manifold learning, graphs are used to represent the manifold. Each node in the graph represents a data point, and edges represent pairwise similarities or affinities between data points. Manifold-aware Neighbor Encoding (MNE) strategy uses this graph representation to capture local neighborhoods and propagates information across the graph to reveal the global structure. Specifically, MNE introduces a novel bidirectional smoothness criterion as a regularization term, ensuring that the learned similarities are smooth across the manifold and helping maintain its local structure in the learned representations. The **visualization** result in Figure 6 highlights that after encoding into the new manifold space, the features exhibit improved manifold properties, which contribute to enhanced retrieval performance.
>
> 2. **Notation Issues**
>
> In line 258, we aim to define a simple diagonal matrix where each diagonal entry is 1. To formalize this, we utilize the delta function, defined as $\delta(i-j)=1$ when $i=j$, and $\delta(i-j)=0$ when $i\neq j$. These two cost matrices can take the same values, but as mentioned in Section 3.2, designing them separately can yield better results. In the revised manuscript, we have refined this definition to improve clarity and the overall readability.
>
> 3. **Dataset Information**
>
> Thanks for your advice. We have provided detailed information about the datasets used and summarized this information in Table 19 of the revised version.
>
> 4. **Effect of Parameter $k_1$**
>
> In our approach, we involve the parameter $k_1$ to determine the activation item for the manifold-aware encoding. The effect of $k_1$ is exhibit in Figure 4. The peak performance is achieved when $k_1$ is around 50, but the performance does not largely vary when $k_1$ exceed 20. The outliers would unavoidably involved in the encoding procedure when $k_1$ increases, but the bidirectional similarity diffusion strategy successfully mitigate their influence, such that the retrieval performance can exhibit a robustness in a wide range. Moreover, we also compare it with the traditional method using the same value in constructing the affinity graph for diffusion, further validate the effectiveness of our manifold-aware neighbor encoding strategy.
>
> 5. **Mistake Highlight**
>
> Sorry for the mistake, we have already corrected it in the revised version.
>
> 6. **More Explanation on the Influence of Outliers**
>
> Actually, to prove that our method can effectively reduce the negative influence of outliers, we involve the **purity metric** to observe the accuracy of true samples in the top-k neighborhood shown in Figure 2, the higher purity represents the more outliers being excluded. Moreover, the retrieval performance which is measured by mean average accuracy metric, quantitatively reflect the ability of reducing the negative impact of outliers in neighborhoods. For the **visual explanation**, we select some instances and exhibit the corresponding weight for constructing manifold-aware neighbor encoding in Figure 5 and provide a t-SNE visualization of MNE feature space in Figure 6. The comparison validates the effectiveness of MNE in strengthening clustering performance and mitigating the effect of outliers. Please refer to the revised version in appendix for more details.

---

### Official Review · Reviewer_GKEp · 2024-11-03

**Soundness:** 3
**Presentation:** 3
**Contribution:** 2
**Rating:** 3
**Confidence:** 5

**Summary:**

This paper presents a Neighbor-aware Geodesic Transportation (NGT) strategy for enhancing the performance of deep clustering and image retrieval tasks. The proposed method consists of three main components: Manifold-aware Neighbor Encoding (MNE), Regularized Barycenter Refinery (RBR), and Geodesic Transportation (GT). By integrating these components, NGT aims to improve the robustness of neighbor identification within the data manifold, thus enhancing the feature learning and clustering results. The method is evaluated on several benchmark datasets, demonstrating superior performance compared to baseline and state-of-the-art methods.

**Strengths:**

(1) Comprehensive Approach: The paper proposes a comprehensive method that integrates multiple strategies (MNE, RBR, GT) to address the challenging task of identifying robust neighbors within the data manifold.
(2) Solid Theoretical Foundation: The method is grounded in solid theoretical principles, with clear explanations of the motivations and methodologies behind each component.
(3) Experimental Validation: The proposed method is thoroughly evaluated on multiple benchmark datasets, demonstrating significant improvements in clustering and retrieval performance compared to baseline and state-of-the-art methods.

**Weaknesses:**

(1) Lack of Novelty: While the proposed method integrates multiple existing techniques, it lacks significant novelty in the underlying algorithms or methodologies. The components (MNE, RBR, GT) are adaptations or combinations of existing approaches rather than entirely new contributions.
(2) Complexity and computational cost: the proposed methodology involves multiple steps and components, which increases the computational complexity and cost compared to simpler baseline methods. Even though the authors performed a time complexity analysis, it still limits its applicability to large-scale datasets or real-time applications.
(3) Limited Scope: The method is specifically tailored for deep clustering and image retrieval tasks and may not be directly applicable to other machine learning or computer vision problems. This limits the broader impact and applicability of the proposed work.
(4) Incremental Improvement: While the proposed method demonstrates improvements over baseline and state-of-the-art methods, the magnitude of these improvements is incremental rather than transformative. This suggests that the method may not offer sufficient advancements to justify its complexity and computational cost in all cases.

**Questions:**

See Weaknesses.

---

> ### Author Response · Authors · 2024-11-24
> **Thank you for your valuable feedbacks.**
>
> 1. **Lack of Novelty**
>
> The **Neighbor-aware Geodesic Transportation (NGT)** introduces unique innovations by systematically combining **Manifold-aware Neighbor Encoding (MNE)**, **Regularized Barycenter Refinery (RBR)**, and **Geodesic Transportation (GT)** to address the challenges of neighborhood refinery. Specifically, **MNE** innovatively encodes global manifold relationships using a bidirectional diffusion process to overcome the limitations of local-only methods. **RBR** incorporates a Wasserstein distance term for robust neighbor integration, mitigating the influence of outliers in a novel way. **GT** introduces geodesic path-based transportation, integrating global geometric and contextual information for refined distance computation. These components represent novel contributions within the context of deep clustering and image retrieval, enabling a systematic framework that advances the state of the art.
>
> 2. **Complexity and Computational Cost**
>
> We acknowledge the computational complexity associated with our approach. However:
>
> - We provide a detailed time complexity analysis (Section 4.3 of the paper), demonstrating that the $\mathcal{O}(n^3)$ complexity of NGT aligns with previously proposed context-based and diffusion-based methods.
> - Our method is **computationally efficient** in practice due to the adoption of iterative optimization strategies and parallelization (e.g., Sinkhorn-Knopp iterations for GT). As shown in Figure 7 and 8, the iterations involved in our methods converge within a few epochs.
> - As noted in Table 6, NGT achieves competitive re-ranking latency on large-scale datasets. Moreover, the coarse-to-fine strategy (explained in Section 4.3) further **reduces computational burden for large-scale applications**. Thus, the observed computational cost is a justified trade-off for the significant performance improvements achieved across diverse datasets.
>
> 3. **Limited Scope**
>
> While the current work focuses on **image retrieval** and **deep clustering**, the core methodology of NGT—manifold encoding, robust barycenter computation, and geodesic transportation—offers generalizable components. Potential applications include: graph-based machine learning tasks, contextual similarity refinement in natural language processing and geometric-aware embedding learning in other domains. We appreciate this feedback and have outlined future work in the conclusion section to explore these broader applications.
>
> 4. **Incremental Improvement**
>
> We respectfully disagree with this assessment. The improvements achieved by NGT are substantial and consistent across multiple benchmarks. For **deep clustering**, NGT demonstrates a **7.8% improvement in NMI** on CIFAR-10 compared to BYOL and outperforms state-of-the-art contrastive and non-contrastive methods (Table 4). For **image retrieval**, NGT achieves **5.7% and 9.8% higher mAP** on ROxf(M) and ROxf(H) compared to the most related ConAff with image feature extracted by R-GeM, highlighting its transformative impact (Tables 1–3). In the appendix (Table 11, 12), we further incorporate NGT with different image retrieval models. These results validate that the proposed framework offers a significant leap in performance, especially considering its robustness to outliers and enhanced neighbor relationships.

---

### Official Review · Reviewer_sbY6 · 2024-11-03

**Soundness:** 2
**Presentation:** 3
**Contribution:** 2
**Rating:** 3
**Confidence:** 3

**Summary:**

This paper investigates how to refine neighbor relationships within the data manifold. Unlike traditional methods that overlook global relationships, this work proposes a novel neighbor-aware geodesic transportation method. Specifically, the paper introduces three modules. The first module is Manifold-aware Neighbor Encoding that project the samples into a nolinear space. The second module is Regularized Barycenter Refinery that integrates local neighbors. The third module is Geodesic Transportation that calculates the shortest path in the affinity graph.

**Strengths:**

a. This paper presents a solution for neighborhood refinement, capturing global relationships to enhance robustness against outliers.

b. The experiments are sufficient and diverse.

**Weaknesses:**

a. Many techniques actually have been proposed and the proposed model seems a combinations of existing methods. For instance, in the Manifold-aware Neighbor Encoding, the authors utilize a Gaussian kernel to construct a manifold-aware space, differing from the traditional use of Euclidean distance to create a Euclidean space. This mapping approach to a nonlinear space is actually a typical technique in spectral clustering. Furthermore, both the neighbor search method in Eq. (3) and the bidirectional similarity diffusion in Eq. (4) are common practices in prior research. Thus, the paper presents limited novel insights.

b. The authors introduce a Regularized Barycenter Refinement module aimed at enhancing robustness against outliers. However, Eq. (6) combines the Euclidean and Wasserstein distances. It’s worth noting that Euclidean distance is inherently sensitive to outliers, making this choice potentially questionable for addressing this issue. It seems that this approach is primarily driven by optimization convenience rather than a robust outlier solution.

c. The third module, as claimed by the authors, uses an existing technique to compute the shortest path in an affinity graph. Consequently, the three primary modules in this work are largely based on established methods.

d. While the authors aim to address robustness to outliers, they do not provide any visual or convincing evidence to support their claims regarding this issue.

e. The authors assert that their approach captures global relationships; however, the techniques employed are still grounded in conventional methods that calculate pairwise affinity weights. Additionally, the lack of visual evidence or further analysis undermines this claim, making it unconvincing.

f. Key experimental details are missing. Firstly, none of the results include standard deviations, which is crucial given the sensitivity of deep learning models to seed selection. Secondly, despite the model’s strict analysis and explicit formulation, the authors do not provide convergence analysis and empirical validation, which is an important omission.

**Questions:**

My questions are in Weaknesses.

---

> ### Author Response · Authors · 2024-11-24
> **Thank you for your valuable feedbacks.**
>
> 1. **Limited Novelty in Techniques and Mapping to Nonlinear Space**
>
> We would like to clarify that our **Manifold-aware Neighbor Encoding (MNE)** strategy is a **novel and effective** approach in mapping original image feature into a nonlinear space, which can encode the essential global and local relationships within the data manifold. Specifically, the **Gaussian kernel** used in MNE is tailored to capture the global structure of the manifold by leveraging a bidirectional similarity diffusion process. This ensures that pairwise relationships are iteratively refined to reflect the intrinsic data geometry, addressing the shortcomings of standard similarity measures. Additionally, the **k-reciprocal neighbor search** (Eq. 3) is designed to guarantee robust and consistent neighborhood definitions, which are crucial for the refinement process in tasks such as re-ranking and deep clustering. These elements form the foundation of a unique encoding framework that systematically captures both local and global dependencies in a way that directly benefits neighborhood refinery tasks. In Figure 6, we employ a t-SNE **visualization of our MNE feature space**, compare with the original image feature, MNE can achieve better clustering performance and reduce the effect of outliers, resulting in a higher retrieval result.
>
> 2. **Euclidean Distance and Outliers**
>
> The concern about Euclidean distance’s sensitivity to outliers in the **Regularized Barycenter Refinement (RBR)** module is valid. However, our choice to include Euclidean distance stems from the following considerations,
>
> - **complementarity**: Euclidean distance captures proximity in the original feature space, while the Wasserstein term accounts for distributional alignment. The Euclidean distance demonstrates high accuracy when retrieving close samples. To effectively address the challenges posed by complex practical datasets, it can be retained as a valuable metric. **The dual objectives ensure a balance between locality and distributional robustness**.
> - **outlier mitigation**: The influence of outliers is mitigated through the **Wasserstein regularization**, which acts as a stabilizing term. Furthermore, our formulation (Eq. 6) includes entropy regularization to facilitate the solution to the optimization problem.
>
> 3. **Use of Existing Techniques in Geodesic Transportation**
>
> We acknowledge that shortest-path computation is a standard technique, but we would like to clarify that the novelty of the **Geodesic Transportation (GT)** module lies in its integration within the NGT framework:
>
> - We choose the geodesic distance to model the **transportation trajectory**. In our NGT framework, it is employed as a cost metric within the optimal transport problem, enabling the incorporation of both global geometric and contextual relationships.
> - The combination of geodesic paths with manifold-aware distributions enhances the refinement of neighborhood relationships in a way that conventional methods do not achieve.
>
> Thus, the value of GT lies in how it synergizes with MNE and RBR to improve neighborhood refinery.
>
> 4. **Visual or Convincing Evidence of Robustness to Outliers**
>
> We appreciate the reviewer’s request for visual evidence to substantiate the robustness claim. To address this, we introduce **neighborhood purity** metric to empirically demonstrate the robustness improvements achieved by the proposed method. Additionally, we select some instances and exhibit the corresponding weight for constructing manifold-aware neighbor encoding in Figure 5 and provide a t-SNE visualization of MNE feature space in Figure 6. The comparison validates the effectiveness of MNE in strengthening clustering performance and mitigating the effect of outliers. Please refer to the revised version in appendix for more details.
>
> 5. **Global Relationship Claims and Lack of Visual Evidence**
>
> We understand the concern regarding global relationship claims and the need for further evidence. To strengthen this aspect, we include **t-SNE visualizations** of the feature space before and after applying our proposed MNE in Figure 6, showcasing enhanced global and local structure alignment.
>
> 6. **Missing Experimental Details**
>
> For more experimental details, we have rerun the experiments across multiple seeds, on CIFAR-10, the impact of random numbers on the results does not exceed 0.2 at most. Moreover, our approach introduces NGT into the BYOL framework to perform deep clustering, which leads to a significant enhancement, e.g., 7.8% improvement in NMI on CIFAR-10. Additionally, we report the convergence analysis in Figures 7 and 8. Both the iterations in MNE and RBR converge quickly within 10 epochs, validating the effectiveness of our method.

---

### Official Review · Reviewer_Q2Kc · 2024-11-04

**Soundness:** 3
**Presentation:** 3
**Contribution:** 3
**Rating:** 5
**Confidence:** 3

**Summary:**

This work proposed a novel Neighbor-aware Geodesic Transportation (NGT) for the neighborhood refinery, which constructs a global-aware distribution for each instance, capturing the intrinsic manifold relationships among all instances.

**Strengths:**

The authors proposed a novel approach for neighborhood refinery that addresses two limitations of previous methods, such as the failure to consider global relationships and not taking into account the negative impact of outliers. The proposed modules Regularized Barycenter Refinery and Geodesic Transportation provide theoretical innovations for solving previous problems. At the same time, a large number of experimental results on re-ranking and deep clustering also provide empirical support. Overall, although the idea is simple and clear, it is indeed effective and provides new insights for subsequent research.

**Weaknesses:**

1) The algorithm involves three modules, which introduces more hyper-parameters and the complexity of each module is the cube of the number of images n.

2) Considering that the proposed method is progressive and contains many steps, an overall algorithm description should be placed in the main text to facilitate the understanding and use of the algorithm.

3) The paper's presentation could be further strengthened to improve readability.

4) Please check for Ps1=a in Line 251, which is not the same as Ps1=ai in line 267. At the same time, double-check for other possible formula errors.

5) In Line 520, Ablations of GT are indicated in Table 4 and is actually shown in Figure 4.

6) GT incorporate the original Euclidean distance and the optimal transport-based distance, and θ is the balance weight. It is encouraged to set θ = 0 to investigate the performance of the method with only the optimal transport-based distance.

**Questions:**

see above.

---

> ### Author Response · Authors · 2024-11-24
> **Thank you for your valuable feedbacks.**
>
> Thanks for your carefully review. To address your concern, here we provide some further explanations below.
>
> 1. **Complexity and Computational Cost**
>
> In our proposed NGT, we modularized our neighborhood refinery method into three distinct modules: Manifold-aware Neighbor Encoding, Regularized Barycenter Refinery and Geodesic Transportation. This modular design not only reduces the implementation difficulty, but also facilitates easy adaptation to different tasks. Moreover, the performance of our algorithm is **robust to variations in most parameters**, enabling optimal results with minimal fine-tuning. In terms of **time complexity**, the theoretical complexity $\mathcal{O}(n^3)$ of our method remains consistent with previous approaches. We have also implemented **extensive optimizations**, including parallelization of key processes, which significantly reduce execution time. As a result, our algorithm achieves **lower deployment latency** compared to most existing methods and can seamlessly integrate into self-supervised learning frameworks. For large-scale datasets, our current strategy focuses on refining the top-ranked sequences, which has been proven to be effective in handling real-world scenarios.
>
> Nevertheless, recalculating the optimal transport distance for each pair of samples leads to substantial computational overhead. To address this, we will focus on developing approximation techniques to further reduce time complexity while maintaining performance in our future work.
>
> 2. **Overall Description and Presentation Issue**
>
> Thanks for your valuable advice, we have placed a brief description of the overall algorithm in the beginning of Section 3 and provided some **visual explanations** in appendix. And we will further improve our writing for clear readability.
>
> 3. **Notation Issue**
>
> In Eq.(6), our initial intention is to present a basic objective function to define the regularized barycenter $a$, with its derivation detailed in Appendix B. To maintain simplicity and consistency with the appendix, we initially omitted subscripts in this definition. While in the subsequent Eq.(7), we modify the formula and introduce subscripts to assign a barycenter $a_i$ to each image. We apologize for any potential confusion caused by the similar notation. In the revised version, **we have consistently included subscripts throughout the main text** to ensure clarity and improve the overall readability.
>
> 4. **Caption Mistake**
>
> Thanks! We have already fixed this issue.
>
> 5. **Analysis of $\theta$ in GT**
>
> Here we replace $\theta=0.05$ with $\theta=0$ and introduce additional comparison intervals with a step size of 0.1. When theta recede to 0,performance is evaluated solely based on the optimal transport distance. Conversely, $\theta=1.0$ represents the case where only the Euclidean distance is used. Since our distance calculation is conducted in an unsupervised manner, it may misclassify individual points as samples from different categories, assigning them larger distances. In contrast, the original Euclidean distance exhibits higher accuracy in distinguishing closely related samples. Through linearly combining these two distance metrics, we can further refine the optimal transport distance to achieve superior performance.
>
> | $\theta$ |  0   | 0.1  | 0.2  | 0.3  | 0.4  | 0.5  | 0.6  | 0.7  | 0.8  | 0.9  | 1.0  |
> | :------: | :--: | :--: | :--: | :--: | :--: | :--: | :--: | :--: | ---- | ---- | ---- |
> | ROxf(M)  | 80.6 | 80.8 | 81.0 | 81.1 | 81.4 | 81.6 | 81.9 | 82.0 | 81.9 | 80.8 | 67.3 |
> | ROxf(H)  | 63.3 | 64.2 | 64.6 | 64.5 | 64.5 | 64.3 | 64.1 | 63.5 | 63.2 | 61.9 | 44.2 |

---

### Note · Authors · 2024-12-28

I have read and agree with the venue's withdrawal policy on behalf of myself and my co-authors.